# Tuning of motor outputs produced by spinal stimulation during voluntary control of torque directions in monkeys

**Miki Kaneshige**[1,2†]**, Kei Obara**[1,3]**, Michiaki Suzuki**[1]**, Toshiki Tazoe**[1]**, Yukio Nishimura**[1]*

[1]Neural Prosthetics Project, Tokyo Metropolitan Institute of Medical Science, Tokyo, Japan; [2]The Japan Society for the Promotion of Science, Tokyo, Japan; [3]Division of Neural Engineering, Graduate School of Medical and Dental Sciences, Niigata University, Niigata, Japan

**Abstract** Spinal stimulation is a promising method to restore motor function after impairment of descending pathways. While paresis, a weakness of voluntary movements driven by surviving descending pathways, can benefit from spinal stimulation, the effects of descending commands on motor outputs produced by spinal stimulation are unclear. Here, we show that descending commands amplify and shape the stimulus-induced muscle responses and torque outputs. During the wrist torque tracking task, spinal stimulation, at a current intensity in the range of balanced excitation and inhibition, over the cervical enlargement facilitated and/or suppressed activities of forelimb muscles. Magnitudes of these effects were dependent on directions of voluntarily produced torque and positively correlated with levels of voluntary muscle activity. Furthermore, the directions of evoked wrist torque corresponded to the directions of voluntarily produced torque. These results suggest that spinal stimulation is beneficial in cases of partial lesion of descending pathways by compensating for reduced descending commands through activation of excitatory and inhibitory synaptic connections to motoneurons.

**\*For correspondence:**
nishimura-yk@igakuken.or.jp

**Present address:** [†]Department of Human Health Sciences, Graduate School of Medicine, Kyoto University, Kyoto, Japan

**Competing interest:** The authors declare that no competing interests exist.

## Editor's evaluation

This study will be of interest to anyone wishing to develop neurotechnologies for restoring motor control following injury. The results convincingly show that spinal stimulation could facilitate or suppress voluntary muscle engagement and joint movement, depending on both the voluntarily evoked activity and the stimulation parameters. This finding is important, as it provides new opportunities for improving stimulation guided neurorehabilitation, particularly in cases of partial lesions.

## Introduction

Electrical stimulation to the spinal cord is a promising method to restore motor function after the impairment of descending pathways through spinal cord injury or stroke. Recent studies have shown that spinal stimulation improves voluntary control of the impaired limb after spinal cord injury in humans (*Angeli et al., 2018*; *Gill et al., 2018*; *Harkema et al., 2011*; *Wagner et al., 2018*) and animals (*Barra et al., 2022*; *Capogrosso et al., 2016*; *Courtine et al., 2009*; *Kasten et al., 2013*; *McPherson et al., 2015*; *Nishimura et al., 2013*; *van den Brand et al., 2012*; *Wenger et al., 2016*). Motor outputs of spinal stimulation have been examined extensively and showed excitatory effects in anesthetized animals (epidural spinal cord stimulation: *Greiner et al., 2021*; intraspinal microstimulation: *Moritz et al., 2007*; *Mushahwar et al., 2004*; *Zimmermann et al., 2011*), spinalized animals

(epidural spinal cord stimulation: *Courtine et al., 2009*; *van den Brand et al., 2012*; *Capogrosso et al., 2016*; *Wenger et al., 2016*; *Barra et al., 2022*; intraspinal microstimulation: *Nishimura et al., 2013*; *Kasten et al., 2013*; *Mushahwar et al., 2004*; *Loeb et al., 1993*; *Tresch and Bizzi, 1999*; *Bizzi et al., 1991*; *Giszter et al., 1993*; *Mussa-Ivaldi et al., 1994*), and humans (epidural spinal cord stimulation: *Angeli et al., 2018*; *Gill et al., 2018*; *Harkema et al., 2011*; *Wagner et al., 2018*). Under these conditions, however, the excitability of motoneurons is too low to observe the effect of inhibitory spinal interneurons on motor outputs, thus, investigations during voluntary movements are necessary.

Paresis is a weakness of voluntary movements caused by partial lesion of descending pathways and is a major symptom in spinal cord injury (*National Spinal Cord Injury Statistical Center, 2021*) and stroke (*Ramnemark et al., 1998*), as well as a major target for therapeutic spinal stimulation. Although individuals with paresis have difficulty controlling their limb movements, they can produce weak muscle activity driven by the preserved descending pathways. In this case, artificial activation of preserved spinal circuits by spinal stimulation can be combined with the influence of preserved descending commands. Although a few studies have examined spinal stimulation in awake animals (*Kato et al., 2020*; *Sharma and Shah, 2021*; *Barra et al., 2022*), the modulation of muscle responses to spinal stimulation by descending commands has not been fully clarified. Descending commands for controlling voluntary limb movements are generated in the motor cortex and activate spinal motoneurons and interneurons. Numerous studies have shown that neural activity in the primary motor cortex represents various movement parameters such as the direction of joint movement (*Caminiti et al., 1990*; *Cisek et al., 2003*; *Crammond and Kalaska, 1996*; *Fu et al., 1993*; *Georgopoulos et al., 1982*; *Georgopoulos et al., 1986*; *Schwartz et al., 1988*) and the amount of muscle activity (*Buys et al., 1986*; *Cheney et al., 1985b*; *Fetz and Cheney, 1980*; *Lemon et al., 1986*). We hypothesized that such parameters during voluntary movement modify the motor outputs evoked by spinal stimulation.

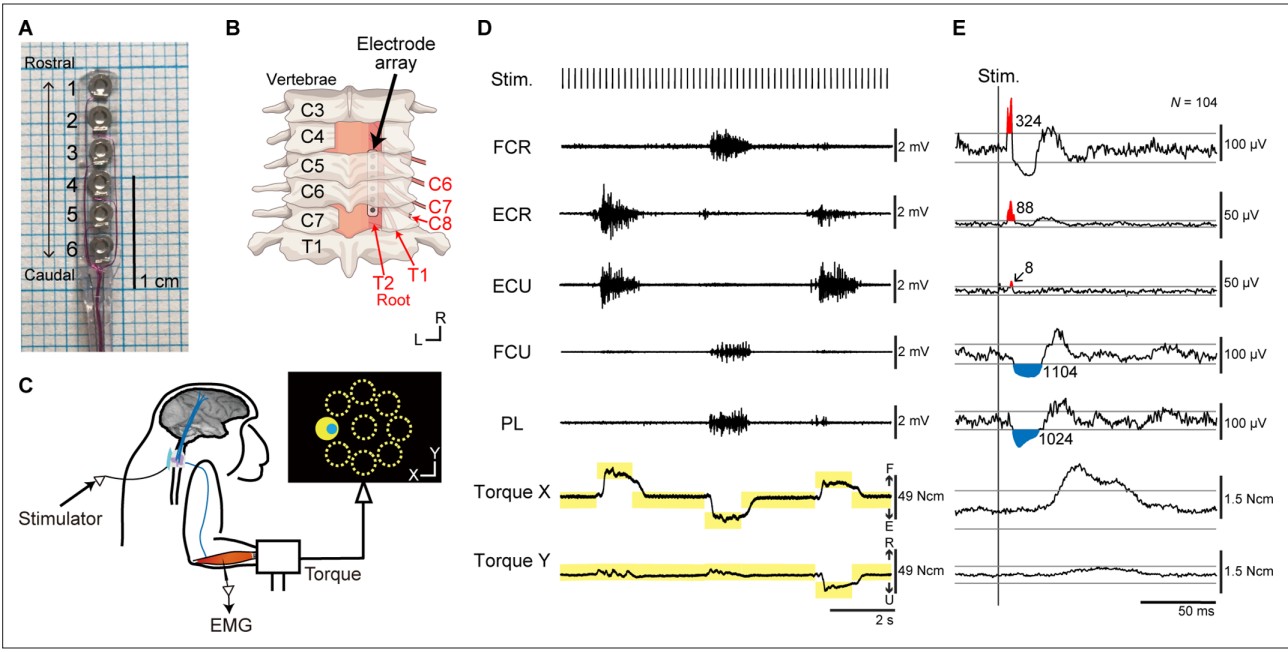

**Figure 1.** Experimental design. (**A**) The platinum electrode array used for subdural spinal stimulation. (**B**) The platinum electrode array was slid into the subdural space from the caudal incision site at the C7 vertebra level and placed over the dorsal-lateral aspect of the C6–T2 spinal segments on right side. (**C**) Spinal stimulation applied during an isometric, eight-target wrist torque tracking task. (**D**) Raw traces of electromyograms (EMGs) and wrist torques during spinal stimulation. One pulse of a biphasic square-wave with a duration of 0.2 ms and an interval of 197 ms was applied at stimulus currents of 110 µA through a single electrode during the task. The yellow rectangles indicate duration and torque of targets. FCR, flexor carpi radialis; ECR, extensor carpi radialis; ECU, extensor carpi ulnaris; FCU, flexor carpi ulnaris; PL, palmaris longus. (**E**) Stimulus-triggered averages (StTAs) of rectified EMGs and torques during the hold period for a peripheral target at stimulus currents of 110 µA through electrode No. 2 (see **A**). Red and blue areas indicate post-stimulus facilitative (Facilitation) and suppressive (Suppression) effects evoked by spinal stimulation, respectively. Numbers give the magnitudes of post-stimulus effects (PStEs) for Facilitation or Suppression (µV·ms). The two gray lines in EMGs or torques represent ± 3 SDs or ± 10 SDs of StTAs calculated during the baseline period (30–10 ms preceding the stimulus trigger pulse), respectively. The data was obtained from monkey W.

**Table 1.** Evoked movements at movement threshold.

| Elec. No. | Monkey H | | | Monkey W | | |
|---|---|---|---|---|---|---|
| | Threshold | Evoked movements | Category of stimulation site | Threshold | Evoked movements | Category of stimulation site |
| 1 | 150 µA | Elbow | Rostral | 120 µA | Elbow | Rostral |
| 2 | 90 µA | Elbow | Rostral | 70 µA | Finger | Caudal |
| 3 | 50 µA | Elbow | Rostral | 40 µA | Thumb | Caudal |
| 4 | 40 µA | Elbow | Rostral | 50 µA | Thumb | Caudal |
| 5 | 30 µA | Thumb | Caudal | 90 µA | Thumb | Caudal |
| 6 | 40 µA | Thumb | Caudal | 80 µA | Wrist | Caudal |
| 7 | 50 µA | Thumb | Caudal | | | |

Here, we investigated the effects of voluntary commands on muscle responses of the upper limb and wrist joint torque induced by subdural spinal stimulation on the cervical enlargement during an eight-directional wrist torque tracking task in monkeys. Results showed that spinal stimulation (150–1350 µA) produced facilitation and/or suppression effects on muscle activities in multiple muscles. The magnitude of these muscle responses showed directional tuning and positively correlated with the level of background muscle activity. Moreover, spinal stimulation boosted torque production in the direction corresponding with the direction of voluntary torque production. These findings suggest that spinal stimulation at an appropriate current is beneficial in a partial lesion of descending pathways to compensate for reduced descending commands by activating excitatory and inhibitory trans-synaptic connections to spinal motoneurons.

## Results

Experiments were performed using two macaque monkeys. A platinum electrode array was chronically implanted in the subdural space over the right-side dorsal rootlets of the cervical enlargement (*Figure 1A, B*). We investigated muscle responses of the right upper limb and wrist torques induced by spinal stimulation to the cervical enlargement under anesthesia and during an isometric, 2D, eight-target, wrist torque tracking task (*Figure 1C–E*).

### Movements induced by spinal stimulation under anesthesia

To characterize spinal sites, we investigated evoked limb movements induced by spinal stimulation under anesthesia. Subdural spinal stimuli consisting of three constant-current, biphasic square-wave pulses of 333 Hz with 0.2 ms duration were delivered to the anesthetized monkeys through a single electrode on the spinal cord. We sampled evoked limb movements for a total of 13 stimulus sites in the two monkeys (seven sites in monkey H and six sites in monkey W). *Table 1* shows the rostro-caudal organization of the evoked movements at movement threshold on the different sites. The electrodes located rostrally tended to induce movements in the proximal arm joints, while caudal electrodes induced movements in distal finger joints. Overall, the evoked movements showed somatotopic representations as described previously (*Kato et al., 2020*; *Sunshine et al., 2013*). Based on these results under anesthesia, we classified the stimulus sites into rostral sites (Elec. No. 1–4 in monkey H and Elec. No. 1 in monkey W), which induced movement in the proximal joints, and caudal sites (Elec. No. 5–7 in monkey H and Elec. No. 2–6 in monkey W), which induced movement in the distal joints (*Table 1*).

### Directional tuning of the evoked muscle responses

Effects of spinal stimulation on target motoneurons can be documented in stimulus-triggered averages (StTAs) of rectified electromyographic (EMG) activity and their response properties during tasks (*Cheney et al., 1985b*; *Cheney and Fetz, 1985a*). Induced muscle responses were assessed by the magnitude of post-stimulus effects (PStEs) in StTAs, compiled while the monkeys were performing an isometric, 2D, eight-target, wrist torque tracking task (*Figure 1C–E*). We examined how PStEs

were modulated by the direction of wrist torques. We sampled PStEs from a total of 1008 muscular conditions (see Materials and methods, Table 1 in *Supplementary file 1*) in 63 experiments (Table 2 in *Supplementary file 1*, stimulus intensity at 20–1600 µA, 7 spinal sites, and 16 muscles in monkey H; stimulus intensity at 10–1700 µA, 6 spinal sites, and 16 muscles in monkey W). *Figure 2A–C* show typical examples of PStEs of rectified EMGs. The PStEs during the entire period of the task (insets on *Figure 2A–C*) showed either post-stimulus facilitative (Facilitation, insets on *Figure 2A and C*) or suppressive effect (Suppression, inset on *Figure 2B*). Spinal stimulation occasionally produced small magnitude of Facilitation during the hold period for the center target where the voluntary wrist torque production was not intended (center panels on *Figure 2A*). However, different magnitudes and/or types of PStEs were observed among the directions of voluntary torques (*Figure 2A–C*). Of the 1008 muscular conditions recorded over all the experiments, 515 muscular conditions in all 16 muscles showed only Facilitation (*Figure 2A*), and 23 muscular conditions in 13 muscles showed only Suppression in all peripheral targets (*Figure 2B*). A total of 469 muscular conditions in 16 muscles changed the type of PStEs, Facilitation or Suppression, depending on the direction (*Figure 2C*). A dominant PStEs type, indicated with a plus symbol (e.g. Facilitation+ and Suppression in *Figure 2C*), was determined by the comparison between the sums of each PStE for the eight target locations. Only one muscular condition in the intrinsic hand muscles showed no response in any of the target locations. Polar plots of *Figure 2A–C* show the magnitudes of Facilitation (red on bottom-left panel), Suppression (blue on bottom-center panel), and background EMG (green on bottom-right panel) during the hold period for the peripheral targets. These magnitudes were significantly tuned in direction, showing the preferred direction (PD, p<0.05, bootstrap). Especially in PStEs of Facilitation, the magnitude of PStEs in the peripheral target close to the PD of background EMG (*Figure 2A*, 270° and 315°) was generally larger compared with that in the center target and smaller in the peripheral target opposite to the PD (*Figure 2A*, 90° and 135°). Significant PDs were observed in the 603 muscular conditions in 16 muscles for Facilitation (Spinal PD of Facilitation), 333 muscular conditions in 16 muscles for Suppression (Spinal PD of Suppression), and 1006 muscular conditions in 16 muscles for background EMG. It should be noted that the Spinal PDs often appear to display similar angles to the PD of background EMG (compare polar plots between bottom-left/bottom-center and bottom-right panels in *Figure 2A–C*).

Population data in *Figure 2D* shows the distributions of significantly tuned PDs. When both Spinal PDs of Facilitation and Suppression were obtained in a single muscular condition (e.g. *Figure 2C*), the Spinal PDs of Facilitation and Suppression were analyzed separately for the population data. The distributions of Spinal PDs of Facilitation (top-left in *Figure 2D*) and Suppression (top-right in *Figure 2D*) were significantly nonuniform and tuned in the ulnar and radial directions, respectively. Similarly, the PD of background EMG (middle panels in *Figure 2D*) also showed nonuniform distributions and were similar with the respective Spinal PD. To omit the effect of directional tuning of the background EMG, we computed the Normalized Spinal PDs for Facilitation and Suppression (bottom panels in *Figure 2D*) by subtracting the PD of background EMG from the Spinal PD. If the PD of background EMG is identical with the Spinal PD, the Normalized Spinal PD should manifest as a distribution centered at 0°. Indeed, Normalized Spinal PDs were significantly tuned around the predicted value of 0° (bottom panels in *Figure 2D*). Therefore, we conclude that muscle responses induced by spinal stimulation were tuned depending on the directions of voluntary torque production and that the PDs of induced muscle responses were identical to the PDs of voluntary activation of the corresponding muscle.

## Effect of current intensity on the evoked muscle responses

Next, we investigated how current intensity affected the magnitudes and directional tuning of the induced muscle responses. *Figure 3A and B* show examples of directional tuning of PStEs at different current intensities. Directional tuning of PStEs was modulated depending on the current intensity. Suppression was dominant at lower currents, while Facilitation was dominant at higher currents (*Figure 3A, B*). The magnitudes of Facilitation increased, and the magnitudes of Suppression decreased with increasing current intensities. In addition, Spinal PDs of Suppression (70 and 180 µA in *Figure 3A*; 70 µA in *Figure 3B*) and Facilitation (1000 µA in *Figure 3A*; 200 and 1000 µA in *Figure 3B*) at lower currents corresponded to the PDs of background EMG, while stimulation at higher currents indicated either no PD (1700 µA in *Figure 3A*) or a significant PD at almost the opposite direction

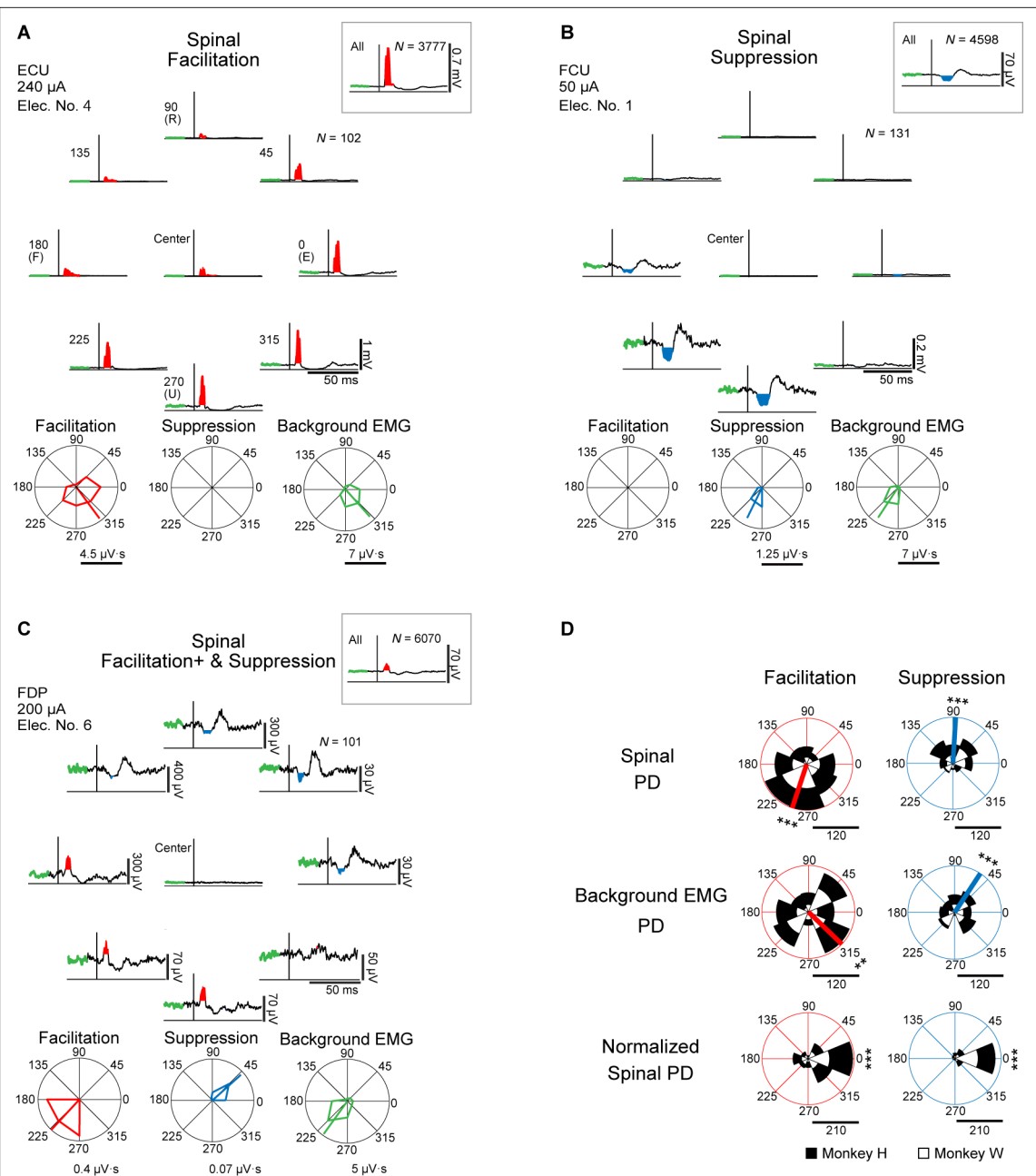

**Figure 2.** Directional tuning of the stimulus-induced muscle responses during wrist torque tracking task. (**A–C**) Muscle responses to spinal stimulation during the hold period for the eight-peripheral (peripheral panels) and the center targets (center of peripheral panels). Insets show the whole period of the task including hold and movement periods. The polar plots display magnitudes of Facilitation (red) and Suppression (blue) effects on post-stimulus effects (PStEs), as well as background electromyograms (EMGs) (green) for a 20 ms pre-stimulus window during the hold period for the eight-peripheral target locations, and the preferred directions (PDs) calculated by vector summation (bootstrap analysis, p<0.05). Typical examples of output type for (**A**) Facilitation (only Facilitation in all targets); (**B**) Suppression (only Suppression in all targets); (**C**) Facilitation+ and Suppression (stimulations induced both effects and larger magnitudes of Facilitation than Suppression). The green thick trace in each stimulus-triggered average (StTA) indicates background EMG activities composing its polar plots. Horizontal bars below the polar plots show the magnitudes of PStEs or background EMGs. ECU, extensor carpi ulnaris; FCU, flexor carpi ulnaris; FDP, flexor digitorum profundus. (**D**) Distributions of the Spinal, background EMG, and Normalized Spinal PDs. Spinal PD (top panels) and background EMG PD (middle panels) show the PDs calculated by the magnitudes of Facilitation or Suppression of PStEs and by the magnitudes of background EMG activity, respectively, during the hold period for the peripheral targets. Normalized Spinal PD (bottom panels) shows angles normalized by subtracting the background EMG PD from the Spinal PD. Red or blue thick lines indicate circular medians and significant nonuniform distributions (Rayleigh test, p<0.05) toward its direction (v-test; *, p<0.05; **, p<0.01; ***, p<0.001). Normalized Spinal PDs

*Figure 2 continued on next page*

*Figure 2 continued*

for Facilitation (bottom-left) and Suppression (bottom-right) show significant nonuniform distributions (Rayleigh test, p<0.05) around 0° (v-test; *, p<0.05; **, p<0.01; ***, p<0.001). Horizontal bars below the polar plots indicate the number of muscular conditions.

The online version of this article includes the following source data for figure 2:

**Source data 1.** Data used to generate polar plots and detailed statistics in *Figure 2D*.

of the PD for background EMG (1700 µA in *Figure 3B*). Thus, stimulus current changes PStEs types ('Facilitation,' 'Suppression,' 'Facilitation+ and Suppression,' and 'Facilitation and Suppression+'), magnitudes, and PD of Facilitation or Suppression.

To characterize the effects of current intensity on the induced muscle responses, we examined the type of PStEs at current intensities of <150 µA, 150–750 µA, 750–1350 µA, and ≥1350 µA (*Figure 3C*). For example, the representative muscles in *Figure 3A and B* showed 'Facilitation and Suppression+' at low-intensity (70 µA in *Figure 3A and B*), whereas these muscles exhibited Facilitation only at high-intensity (1700 µA in *Figure 3A and B*). In addition to the representative data, population analysis showed mainly 'Facilitation+ and Suppression' or 'Facilitation and Suppression+' at lower currents. The percentage of 'Facilitation' increased and the percentage of other types decreased as the stimulus current increased (*Figure 3C*). We next quantified the effect of current intensity on the magnitudes of PStEs, which were defined as the sum of each PStE for the eight targets. The magnitudes of Facilitation (left panel in *Figure 3D*) increased as the current intensity increased. The magnitudes of Suppression at higher currents tended to decrease compared to those at lower current (right panel in *Figure 3D*). Thus, the current intensity tuned the type and magnitude of PStEs.

We further characterized the effects of current intensity on distributions of the Spinal PDs (*Figure 3E and F*). The Spinal PDs for Facilitation and Suppression at lower current (<1350 µA) exhibited significant nonuniform distributions toward ulnar and radial directions, respectively (*Figure 3E*). The Normalized Spinal PDs showed commonly nonuniform distributions around 0° (*Figure 3F*). In contrast, Spinal PD for Facilitation at higher currents (≥1350 µA) showed uniform distributions (bottom-left panel in *Figure 3E*), and the Normalized Spinal PD for Facilitation exhibited the opposite to PD of background EMG (bottom-left panel in *Figure 3F*). These results, in which low currents induced the Spinal PD similar to the PD of background EMG and large magnitudes of stimulus effects were induced in the direction of large magnitudes of background EMG, while high currents produced the Spinal PD opposite to the PD of background EMG, correspond to the typical examples of *Figure 3A and B*. Thus, relations between the Spinal PD and the PD of background EMG changed by current intensity, implying that recruited neural elements depend on current intensity.

## Effect of the stimulus sites on the evoked muscle responses

To illustrate the effects of stimulus sites on the induced muscle responses during the task, we classified the stimulus sites into rostral sites and caudal sites based on the evoked movements under anesthesia as described above (*Table 1* and *Figure 4A*). Furthermore, since the motor nucleus of each muscle is distributed differently within the spinal cord, the recorded muscles were divided into two groups, rostrally innervated muscles and caudally innervated muscles, based on previous electrophysiological and anatomical evidence (*Fritz et al., 1986*; *Fritz et al., 1982*; *Jenny and Inukai, 1983*; *Schieber et al., 1997*; *Schirmer et al., 2011*; *Figure 4—figure supplement 1*). Biceps brachii (BB), brachioradialis (BR), pronator teres (PT), flexor carpi radialis (FCR), and extensor carpi radialis (ECR) were categorized as rostrally innervated muscles. Triceps brachii (Triceps), palmaris longus (PL), flexor carpi ulnaris (FCU), extensor carpi ulnaris (ECU), flexor digitorum superficialis (FDS), flexor digitorum profundus (FDP), extensor digitorum communis (EDC), extensor digitorum 4 and 5 (ED4, 5), abductor pollicis longus (APL), first adductor pollicis (ADP), and abductor digiti minimi (ADM) were categorized as caudally innervated muscles.

*Figure 4B and C* show examples of directional tuning of PStEs induced from two different rostral (Elec. No. 1) and caudal (Elec. No. 5) sites at the same stimulus intensity during the task. In the rostrally innervated muscle (BB), stimulation through a rostral site (Elec. No. 1 indicated by yellow in *Figure 4A*) evoked Facilitation, and the Spinal PD was toward the radial direction, which was similar to the PD of background EMG (*Figure 4B*, top). Stimulation through a caudal site (Elec. No. 5 indicated by green in *Figure 4A*) evoked the effect of Facilitation and Suppression+ (*Figure 4B*, bottom), and Spinal PD

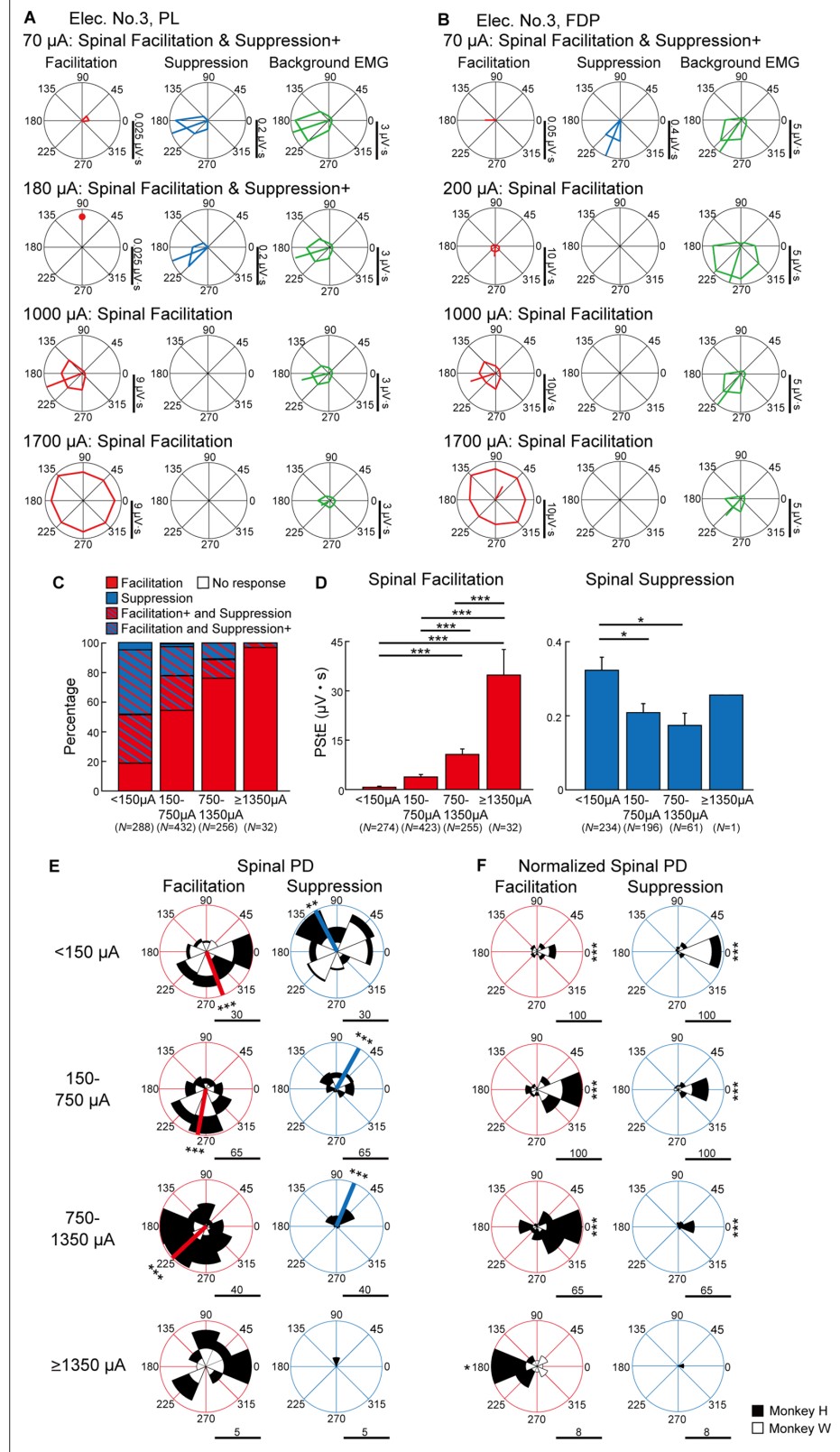

**Figure 3.** Effect of current intensity on directional tuning of the stimulus-induced muscle responses. (**A and B**) Polar plots of the post-stimulus effects (PStEs) for Facilitation, Suppression, and background electromyogram (EMG) at four different current intensities during the hold period for the eight peripheral targets. Vertical bars next to the polar plots show the magnitudes of PStEs or background EMGs. (**A**) Responses in the palmaris longus (PL)

*Figure 3 continued on next page*

*Figure 3 continued*

muscle at current intensities of 70 µA, 180 µA, 1000 µA, and 1700 µA through Elec. No. 3. (**B**) Responses in the flexor digitorum profundus (FDP) muscle by stimulation at current intensities of 70 µA, 200 µA, 1000 µA, and 1700 µA through Elec. No. 3. (**C**) Effect of current intensity on output type for PStEs. Facilitation, only Facilitation in all targets; Suppression, only Suppression in all targets; Facilitation+ and Suppression, stimulations induced both PStEs, and larger magnitudes of Facilitation than Suppression; Facilitation and Suppression+, stimulations induced both PStEs and larger magnitudes of Suppression than Facilitation; No response, no PStEs in all targets. (**D**) Mean values and SEs for the magnitudes of PStEs calculated in each current intensity. Statistics: one-way factorial ANOVA with Tukey-Kramer correction for post hoc multiple comparison (*, p<0.05; **, p<0.01; ***, p<0.001). (**E and F**) Population data of effect of current intensity on (**E**) Spinal PD and (**F**) Normalized Spinal PD. Red or blue thick lines indicate circular medians and significant nonuniform distributions (Rayleigh test, p<0.05) toward its direction (v-test; *, p<0.05; **, p<0.01; ***, p<0.001). Normalized Spinal PDs for Facilitation and Suppression evoked by lower (<1350 µA) intensity stimulation were significantly nonuniform (Rayleigh test, p<0.05) around 0° (v-test; *, p<0.05; **, p<0.01; ***, p<0.001), whereas Normalized Spinal PDs for Facilitation evoked by high intensity stimulation (≥1350 µA) was significantly nonuniform (Rayleigh test, p<0.05) around 180° (v-test; *, p<0.05; **, p<0.01; ***, p<0.001). Horizontal bars below each polar plot show the number of muscular conditions.

The online version of this article includes the following source data for figure 3:

**Source data 1.** Data used to generate bar plots and detailed statistics in *Figure 3D*.

**Source data 2.** Data used to generate polar plots and detailed statistics in *Figure 3E*.

**Source data 3.** Data used to generate polar plots and detailed statistics in *Figure 3F*.

of Suppression was similar to the PD of background EMG, while Spinal PD of Facilitation was opposite to the PD of background EMG. In the caudally innervated muscles (FDP), stimulation through a rostral site (yellow, Elec. No. 1) or a caudal site (green, Elec. No. 5) evoked Facilitation only, and the Spinal PDs of Facilitation were close to the PDs of background EMG (*Figure 4C*).

Population data in *Figure 4D* shows the effect of stimulus site on the type of PStEs. Regardless of stimulus site, the majority of the PStEs were 'Facilitation' or 'Facilitation and Suppression' in both rostrally (left panel) and caudally (right panel) innervated muscles. Population data in *Figure 4E and F* compares the magnitudes of PStEs between rostral and caudal sites. Stimulation at rostral sites exhibited larger magnitudes of Facilitation effects into the rostrally innervated muscles than stimulation at caudal sites (*Figure 4E*, left panel). Similarly, in the caudally innervated muscles, stimulation at caudal sites produced larger magnitudes of PStEs on both Facilitation and Suppression than stimulation at rostral sites (*Figure 4F*).

We further investigated the effects of stimulus sites on the Spinal PD. Regardless of stimulus sites, almost all Spinal PDs tuned to directions corresponding to the PDs of background EMG (first and fourth row panels in *Figure 4G and H*). That is, the Normalized Spinal PDs showed nonuniform distributions around 0° (third and sixth row panels in *Figure 4G and H*). Thus, regardless of the distance from the stimulus site to the motor nucleus for each muscle, Spinal PDs corresponded with the PDs of background EMG.

On stimulation at caudal sites, Spinal PD of Facilitation in rostrally innervated muscles was opposite to the PD of background EMG (compare between top-left panel and middle-left panel in caudal site of *Figure 4G*), and the Normalized Spinal PDs for Facilitation exhibited nonuniform distributions around 180° (*Figure 4G*, bottom-left panel). On the other hand, stimulation at caudal sites produced Spinal PDs for Suppression tuned with radial directions into the rostrally innervated muscles (top-right panel in caudal site of *Figure 4G*), which was similar to the PD for background EMG (middle-right panel in caudal site of *Figure 4G*).

## Effect of background EMG on the evoked muscle responses

We found that induced muscle responses were tuned by the directions of voluntary torque and that Spinal PD corresponded to the PD of background EMG (*Figure 2*). Since Spinal PD and the PD of background EMG were identical, we hypothesized that induced muscle responses depend on the excitability of the motoneuron pool. To examine this issue, we investigated the relationship between the magnitudes of the PStEs and background EMGs. EMG activity obtained from the eight-directional task was divided into five different levels of background EMGs for analysis, and PStEs were shown based on the magnitudes of background EMG (*Figure 5A–C*). Left insets and gray dots in right panels

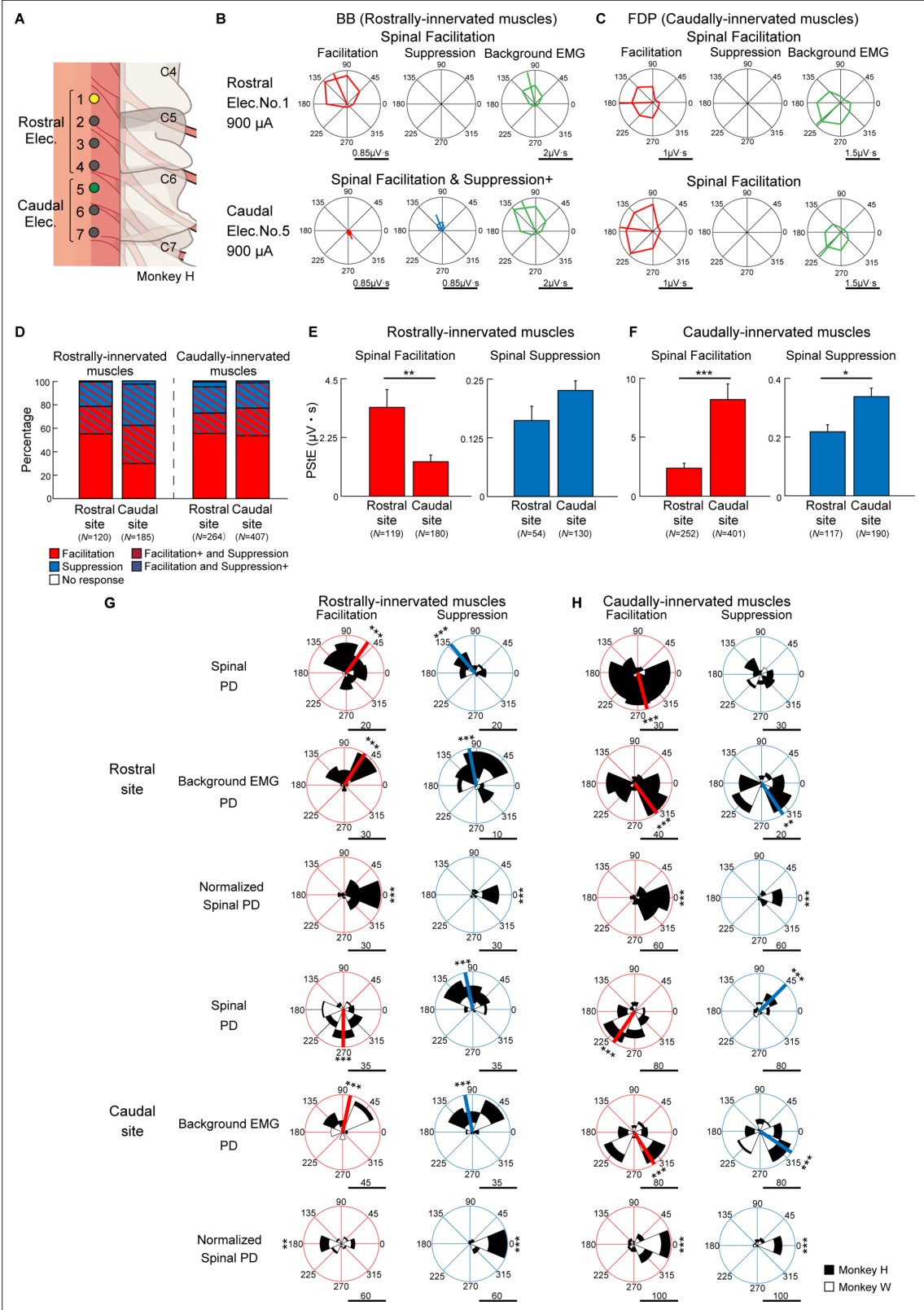

**Figure 4.** Effect of stimulus site on directional tuning of the stimulus-induced muscle responses. (**A**) Locations of electrodes on the cervical cord. Yellow and green sites correspond with rostral (top rows) and caudal (bottom rows) electrodes on (**B**) and (**C**), respectively. (**B and C**) Polar plots of the muscle responses for Facilitation, Suppression, and background electromyogram (EMG) by stimulations from the rostral electrode and caudal electrode during the hold period for the eight peripheral targets. These muscle responses are obtained from the (**B**) biceps brachii (BB) muscle and (**C**) flexor digitorum

*Figure 4 continued on next page*

*Figure 4 continued*

profundus (FDP) muscle innervated by motoneurons located in the rostral and caudal cervical cord, respectively. Horizontal bars below the polar plots show the magnitudes of post-stimulus effects (PStEs) or background EMGs. (**D**) Effect of stimulations at rostral and caudal sites on PStE types in the rostrally and caudally innervated muscles. The color-coded representations are the same as in *Figure 3C*. (**E and F**) Mean values and SEs for the magnitudes of PStEs from rostral or caudal stimulus sites into (**E**) rostrally innervated muscles or (**F**) caudally innervated muscles (two-sided unpaired t-test; *, p<0.05; **, p<0.01; ***, p<0.001). (**G and H**) Distributions of the PDs for Facilitation and Suppression from (**G**) rostrally innervated muscles or (**H**) caudally innervated muscles by stimulation at rostral (top) or caudal (bottom) sites. The Normalized Spinal PDs for Facilitation and Suppression are significantly tuned (Rayleigh test, p<0.05) around 0° (v-test; *, p<0.05; **, p<0.01; ***, p<0.001), except for the cases of Facilitation in the rostrally innervated muscles through the stimulation at caudal sites (bottom-left panel of [**G**]). Red or blue thick lines indicate circular medians and significant nonuniform distributions (Rayleigh test, p<0.05) toward its direction (v-test; *, p<0.05; **, p<0.01; ***, p<0.001). Horizontal bars below each polar plot show the number of muscular conditions. Because higher currents of ≥1350 µA were administered using only caudal electrodes, the data obtained at high-intensity stimulation was excluded from the population analyses in Figure 4 to allow for fair comparison.

The online version of this article includes the following source data and figure supplement(s) for figure 4:

**Source data 1.** Data used to generate bar plots and detailed statistics in *Figure 4E*.

**Source data 2.** Data used to generate bar plots and detailed statistics in *Figure 4F*.

**Source data 3.** Data used to generate polar plots and detailed statistics in *Figure 4G*.

**Source data 4.** Data used to generate polar plots and detailed statistics in *Figure 4H*.

**Figure supplement 1.** Definition of stimulus sites and muscles.

(*Figure 5A–C*) show the PStEs and background EMGs during hold period for the center target. Regardless of the type of PStEs, the magnitude of PStEs increased as background EMG increased (right panels in *Figure 5A–C*, two-sided Pearson's correlation, *Figure 5A*, r=0.950, p=0.01; *Figure 5B*, r=0.997, p=2.00 × $10^{-4}$; *Figure 5C*, r=0.999, p=4.63 × $10^{-5}$), and most muscular conditions showed significant positive correlations between the magnitudes of PStEs and background EMGs (hatched bars in *Figure 5D*). These results indicate that the magnitudes of the induced muscle responses altered by the direction of voluntary torques are tuned by the excitability of spinal motoneurons.

We found that Spinal PDs were changed by current intensity, as well as PDs of background EMG (*Figure 3*). To elucidate how current intensity affects the relationship between muscle responses and the excitability of the motoneuron pool, we investigated the effect of current intensity on the relation between the magnitudes of PStEs and background EMGs (*Figure 6A–D*). Lower current (70 µA) produced Suppression and showed a strong positive correlation between the magnitudes of PStEs and background EMGs (*Figure 6A*, two-sided Pearson's correlation, r=0.995, p=4.00 × $10^{-4}$). On the other hand, medium (150 µA and 1000 µA) and higher (1700 µA) current stimulations induced Facilitation and resulted in a saturation of Facilitation, with no significant correlations (two-sided Pearson's correlation, *Figure 6B*, r=0.862, p=0.06; *Figure 6C*, r=0.611, p=0.27; *Figure 6D*, r=–0.788, p=0.11). Moreover, higher currents at 1700 µA (*Figure 6D*) tended to reduce the magnitudes of Facilitation on higher background EMG. PStEs during the hold period for the center target increased as current intensity increased, showing a simple input-output property of stimulus-indued muscle responses ('center target,' insets on *Figure 6A–D*). In general, including the hold period for the center target, the magnitudes of PStEs at low stimulus currents were linearly increased depending on the magnitudes of background EMGs (*Figure 5A–C* and *Figure 6A*). However, the magnitudes of PStEs of Facilitation at medium currents were often larger during hold period for the center target (*Figure 6B, C* insets) compared to that during voluntary torque production even though the magnitude of background EMG was identical between them (*Figure 6B and C*, rightmost panels).

*Figure 6E* shows the population data of the correlation coefficients for the relationship between the magnitudes of PStEs and background EMGs at different stimulus currents. Most muscular conditions at lower currents (<1350 µA) showed a significant positive relationship in all PStEs types (first to third row panels in *Figure 6E*). In contrast, the distribution of correlation coefficient at higher currents (≥1350 µA) showed a uniform distribution that included a negative relationship between the magnitudes of PStEs and background EMGs (bottom-left panel in *Figure 6E*). Consequently, these results indicate that stimulus-induced muscle responses at lower currents were amplified depending on the amount of descending commands to spinal motoneurons. On the other hand, stimulus-induced muscle responses at higher current were attenuated on higher background EMG. These findings clarify the observation that Spinal PD corresponds to the PD for background EMG at lower currents but not at higher currents.

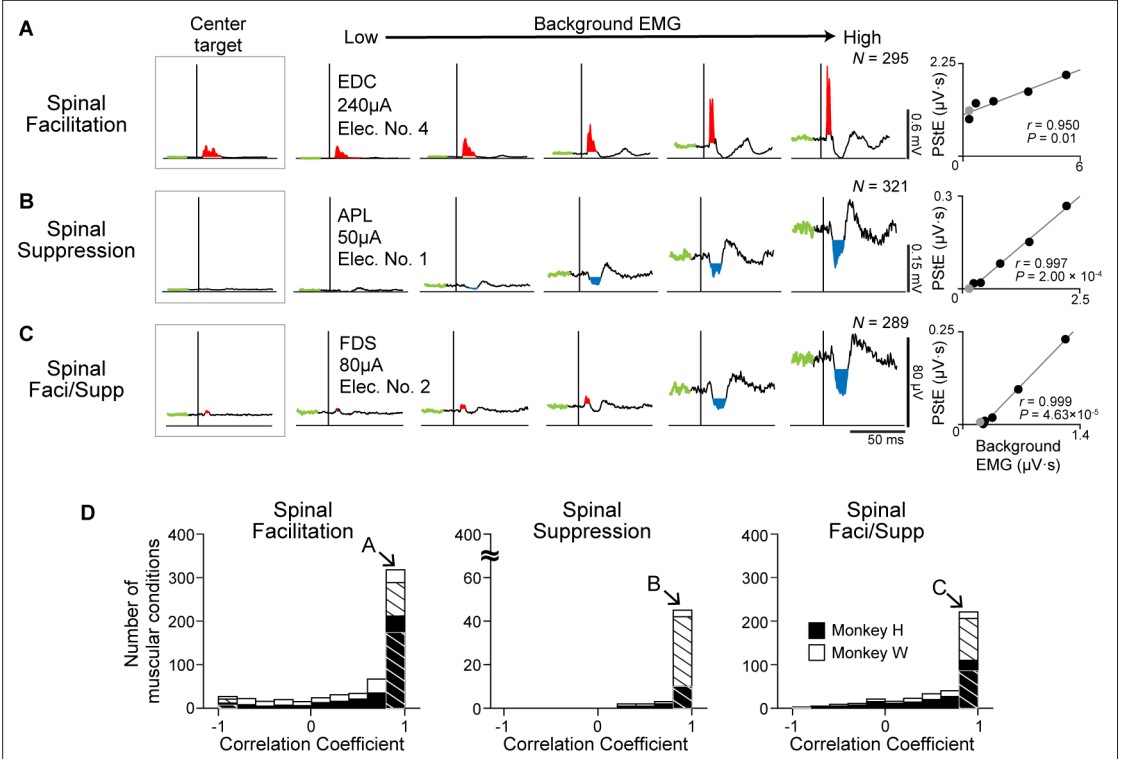

**Figure 5.** Effect of background electromyogram (EMG) on the stimulus-induced muscle responses. (**A–C**) Examples of background EMG-dependent modulations for the post-stimulus effects (PStEs). Representative examples of PStEs for (**A**) Facilitation only, (**B**) Suppression only, and (**C**) both Facilitation and Suppression (Faci/Supp). The leftmost insets show PStEs during the hold period for the center target. The rightmost panels for each muscular condition show two-sided Pearson's correlation coefficients between the magnitudes of background EMGs and PStEs. Gray dots in right panels indicate the result during the hold period for the center target that were not included for the correlation analyses. (**D**) Population data of the correlation coefficients between the magnitudes of background EMGs and PStEs. Correlation coefficients are categorized by output type of PStEs altered depending on the magnitudes of background EMGs, which indicated only Facilitation (Facilitation, left), only Suppression (Suppression, center), and both the Facilitation and Suppression (Faci/Supp, right). There are strong positive correlations between the magnitudes of background EMGs and any type of PStEs (i.e. Facilitation, Suppression, and Faci/Supp). Hatched bars indicate the number of muscular conditions showing significant correlation between the magnitudes of PStEs and background EMGs (two-sided Pearson's correlation test, p<0.05). Unhatched bars show the conditions with no statistical significance. The letter shown in each arrow identifies the muscular conditions in **A, B, and C**, respectively.

## Tuning of evoked wrist torque

Finally, we investigated how voluntary commands modify induced wrist torque. *Figure 7B–D* show typical examples of the trajectories depicted by PStEs of torques (Evoked Torque, gray, top-center of the peripheral panels, and inner peripheral panels) and the StTAs of all muscles (bottom-center of the peripheral panels and outer peripheral panels) during the hold period for each center and peripheral target. The length and direction of arrows represent magnitudes and directions of Evoked Torque (a value near each arrow in the inner peripheral panels in *Figure 7B–D* indicates the direction of Evoked Torque), respectively, and, together, indicate that spinal stimulation induced significant Evoked Torque.

Spinal stimulation at 110 µA tended to induce Suppression effects on muscles with higher background EMG (outer peripheral panels in *Figure 7B* and *Figure 7—figure supplement 1A*). The directions of Evoked Torque were opposite to the directions of voluntary torque and were converged to the center target where the wrist is relaxed (inner peripheral panels in *Figure 7B*). To investigate the relation between the directions of the Evoked Torque and the voluntary torque, Normalized Torque was computed by subtracting the direction of voluntary torque from that of Evoked Torque for the peripheral targets. Normalized Torque at 110 µA was approximately 180°, indicating that the direction of Evoked Torque was opposite to the directions of voluntary torque (inset on *Figure 7B*).

In another case, spinal stimulation at 300 µA mainly induced Facilitation effects on muscles with higher background EMG (outer peripheral panels in *Figure 7C* and *Figure 7—figure supplement*

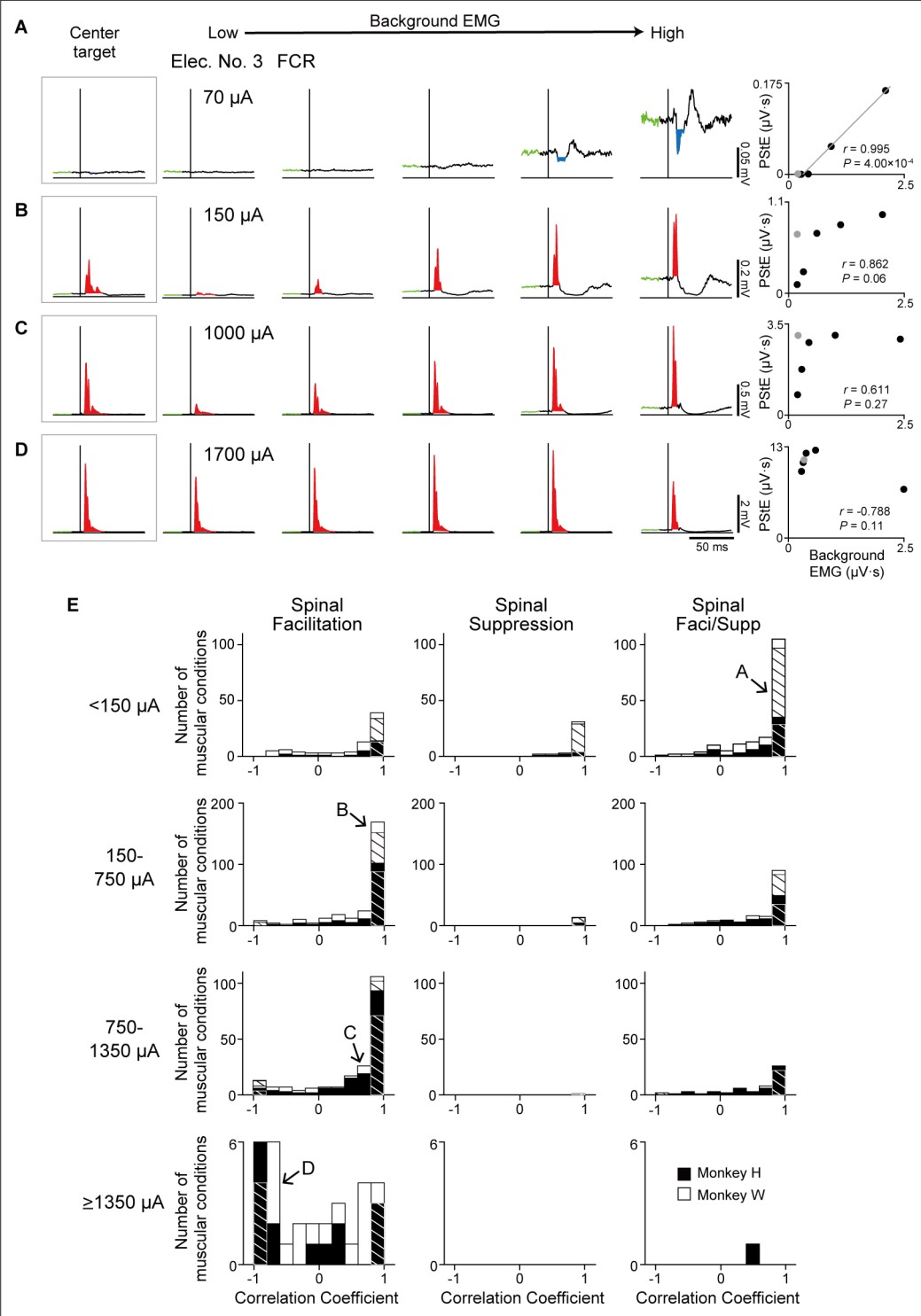

**Figure 6.** Effect of current intensity on the relationship between the stimulus-induced muscle responses and background electromyograms (EMGs). (**A–D**) Examples of stimulus-intensity-dependent modulation of post-stimulus effects (PStEs) on background EMGs. Stimulus-triggered averages (StTAs) by stimulation at stimulus intensities of (**A**) 70 μA, (**B**) 150 μA, (**C**) 1000 μA, and (**D**) 1700 μA were obtained from the flexor carpi radialis (FCR) muscle through Elec. No. 3 of monkey W. The leftmost insets show PStEs during hold period for the center target intended to relax the wrist. The rightmost panels indicate two-sided Pearson's correlation coefficients between the magnitudes of background EMGs and PStEs. Gray dots in right panels indicate the result during hold period for the center target that were not included for the correlation analyses. (**E**) The distributions of two-sided Pearson's

*Figure 6 continued on next page*

*Figure 6 continued*

correlation coefficients between the magnitudes of background EMGs and PStEs at stimulus intensities of <150 μA, 150–750 μA, 750–1350 μA, and ≥1350 μA. Correlation coefficient for each muscular condition was categorized as 'Facilitation only' (Facilitation, left panels), 'Suppression only' (Suppression, center panels), and 'Facilitation and Suppression' (Faci/Supp, right panels) according to the output type of PStEs. The magnitudes of PStEs at lower current stimulation show positive correlation with those of background EMGs, whereas the magnitudes of PStEs at higher current stimulation exhibit negative correlation with those of background EMGs. Hatched and unhatched bars indicate the number of muscular conditions as in *Figure 5D*. The letter shown in each arrow indicates the muscular conditions in **A, B, C, and D**, respectively.

*1B*), and the directions of the Evoked Torque were similar to the directions of voluntary torque independent of the direction of the Evoked Torque at the center target (center and inner peripheral panels in *Figure 7C*).

Stimulation at 1700 μA exhibited large magnitudes of Facilitation in all muscles for all targets (outer peripheral panels in *Figure 7D* and *Figure 7—figure supplement 1*) and the Evoked Torques displayed ulnar-flexion directions regardless of the presence/absence or the direction of voluntary torque (center and inner peripheral panels in *Figure 7D*). During the eight-directional torque task, the monkeys properly engaged each muscle as agonist (*Figure 7—figure supplement 1*). We found the antagonistic voluntary contraction was quite rare or mostly nondominant even during high-intensity electrical stimulation. There was a tendency that the magnitude of PStEs was stronger in agonists and weaker in antagonists at low and medium currents (*Figure 7—figure supplement 1A, B*). On the other hand, stimulation at high currents tended to induce large magnitudes of facilitation effects for all targets irrespective of agonist and antagonists (*Figure 7—figure supplement 1C*).

Population data showed that the magnitudes of Evoked Torque for the peripheral targets increased as the current intensity increased (*Figure 7E*). Lower currents (<150 μA) exhibited uniform distribution in the directions of the Evoked Torque (top-left panel in *Figure 7F*) and were opposite to the directions of voluntary torque (top-right panel in *Figure 7F*). Medium currents (150–1350 μA) induced Evoked Torque predominately toward the ulnar-flexor direction (second and third-left panels in *Figure 7F*), and the directions of Normalized Torque corresponded to the direction of voluntary torque (second- and third-right panels in *Figure 7F*). Since high currents (≥1350 μA) displayed Evoked Torques only toward the ulnar-flexion direction regardless of the direction of voluntary torque (bottom-left panel in *Figure 7F*), the direction of Normalized Torques showed uniform distributions (bottom-right panel in *Figure 7F*). Accordingly, magnitudes and directions of stimulus-induced wrist torques were modulated according to the direction of voluntary torque and current intensity. At medium current (150–1350 μA), spinal stimulation boosted torque outputs in the same direction as on-going voluntary torque production.

## Onset latency of evoked muscle responses altered by different current intensities

We found that stimulations at different current intensities induced different types of evoked muscle responses (*Figure 3C*), magnitudes of evoked muscle responses (*Figure 3D*), relations between muscle responses and background EMGs (*Figure 6E*), and directions and magnitudes of evoked torque (*Figure 7F*). However, it is unclear whether different current intensities of stimulations recruit different pathways. To answer this question, we examined the onset latencies of PStEs. *Figure 8A and B* exhibit typical examples showing effect of current intensity on onset latencies of PStEs. Regardless of whether the current intensity changed the type of PStEs (*Figure 8A and B*), the onset latency of the PStEs was found to shorten as the current increased.

We further examined effect of directions of voluntary torque and onset latencies of PStEs. *Figure 8C* shows PStEs in the PL muscle at four different current intensities during the hold periods for the eight-peripheral targets. In this example, stimulation at lower current (70 μA) clearly changed onset latencies of PStEs depending on directions of voluntary torque, while onset latencies of PStEs at higher currents (1000 and 1700 μA) were similar among different directions. Population data in *Figure 8D* shows the distributions of onset latencies of PStEs at four different current intensities during the hold period for the peripheral targets. The median onset latency of Facilitation effects was shorter than that of Suppression effects at all current intensities. In addition, the median onset latencies of Facilitation

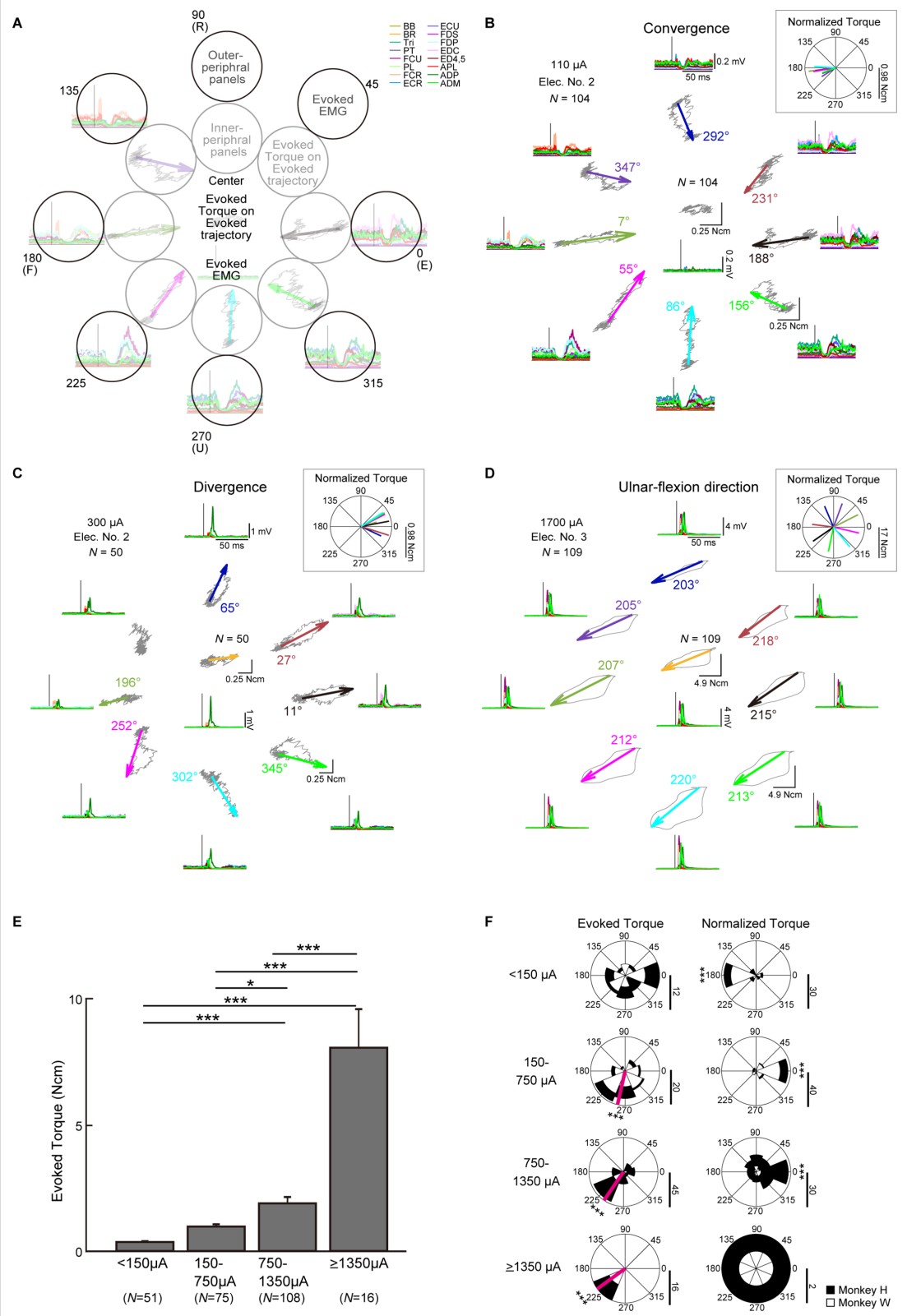

**Figure 7.** Directional tuning of the stimulus-induced wrist torque. (**A**) The structure of **B–D**. The center panel shows evoked torque trajectory about wrist joint (top) and evoked electromyogram (EMG) (bottom) during the hold period for the center target. Inner- (gray circles) and outer-peripheral panels (black circles) indicate evoked torque trajectory and evoked EMG on the eight-peripheral targets, respectively. (**B–D**) Stimulus-triggered averages (StTAs) of rectified EMGs (outer peripheral panels and center-bottom panel) and StTAs of wrist torque trajectories (inner peripheral panels

*Figure 7 continued on next page*

*Figure 7 continued*

and center-top panel). The length of arrows and the direction of arrowhead in inner peripheral panels express the magnitudes and directions of statistically significant evoked torques (Evoked Torque), respectively (see 'Materials and methods'). The absence of an arrow for the radial-flexion location of (**C**) indicates no statistically significant Evoked Torque. Color-coded numbers near each arrow indicate the direction of Evoked Torque. Normalized Torque (insets of **B-D**) was exhibited by subtracting the direction of voluntary torque productions from the direction of Evoked Torque for the eight-peripheral targets. Vertical bars of polar plots (insets of **B-D**) display the magnitudes of Normalized Torque. Typical examples of Evoked Torque type. (**B**) Convergence, stimulation at 110 µA induced torque toward the center in the eight-peripheral targets. (**C**) Divergence, stimulation at 300 µA induced outward torque in the eight-peripheral targets. (**D**) Ulnar-flexion direction, stimulation at 1700 µA induced torque toward the ulnar-flexion direction in the eight-peripheral targets. (**E**) Mean values and SEs for the magnitudes of Evoked Torque for the eight-peripheral targets calculated in each current intensity. Statistics: one-way ANOVA with Tukey-Kramer correction for post hoc multiple comparison (*, $p < 0.05$; **, $p < 0.01$; ***, $p < 0.001$). (**F**) Distributions for the directions of the Evoked Torque (left panels) and the Normalized Torque (right panels) for the eight-peripheral targets. Pink lines indicate circular medians and significant nonuniform distributions (Rayleigh test, $p < 0.05$) toward its direction (v-test; *, $p < 0.05$; **, $p < 0.01$; ***, $p < 0.001$). The Normalized Torque at <150 µA and 150–1350 µA significantly tuned (Rayleigh test, $p < 0.05$) around 0 and 180° (v-test; *, $p < 0.05$; **, $p < 0.01$; ***, $p < 0.001$), respectively. Vertical bars below the polar plots indicate the number of muscular conditions.

The online version of this article includes the following source data and figure supplement(s) for figure 7:

**Source data 1.** Data used to generate bar plots and detailed statistics in *Figure 7E*.

**Source data 2.** Data used to generate polar plots and detailed statistics in *Figure 7F*.

**Figure supplement 1.** Subdural spinal stimulation simultaneously evoked facilitative and suppressive effects in multiple muscles and activated synergistic muscle groups.

effects became shorter as stimulus currents increased, while those of Suppression effects were similar among the four stimulus currents. These current-dependent changes in onset latency indicate that the stimulus current affects recruited neural elements.

## Discussion

This study aimed to clarify the effects of voluntary commands on muscle responses and wrist torques induced by spinal stimulation in monkeys. During voluntary torque productions at the wrist in eight different directions and 45° apart, spinal stimulation over the C6-T2 region produced facilitation and/or suppression effects on muscle activities in multiple muscles of the upper limb. The magnitude of these muscle responses was tuned by voluntary commands that controlled the direction of torque production and the level of background muscle activity. The stimulus-induced muscle responses were also associated with current intensity. At lower currents (<1350 µA), the PDs of muscle responses corresponded to those of background muscle activity. The underlying mechanisms were explained by the observation that the magnitudes of muscle responses positively correlated with the levels of muscle activity, reflecting the level of descending commands to spinal motoneurons. This relationship disappeared at higher currents (≥1350 µA). Moreover, the induced wrist torques were modulated by directions of voluntary torque and stimulus currents. Appropriate currents (150–1350 µA) evoked torques toward the same direction as voluntary torque production. Thus, spinal stimulation at balanced currents of excitation and inhibition in spinal circuits boosts torque production in directions corresponding to the direction of voluntary torque production.

### Current-dependent activation of neural circuits

The type of induced muscle responses and the direction of evoked torques during the task changed depending on the stimulus current (*Figures 3 and 7*). The number of muscles showing suppression effects and its magnitude decreased, while those of facilitation effects increased as current intensity increased (*Figure 3*). Similarly, lower currents suppressed voluntary torques, while medium currents boosted torques toward the same direction as voluntary torque production (*Figure 7*). Onset latencies of evoked muscle responses were shorter with increasing current intensities (*Figure 8A, B and D*). Such current-dependent effects indicate that the recruited neural elements changed as the stimulus current increased during voluntary torque production.

Since the subdural arrays were placed over the dorsal rootlets (*Figure 1B*), electrical currents are likely to first drive the afferent fibers adjacent to the stimulus sites, indicating that a major component of stimulus effect could be driven by spinal reflex via large diameter and low threshold afferent fibers such as Ia, Ib, and cutaneous afferents. Suppression effects are mediated, at least in part, by

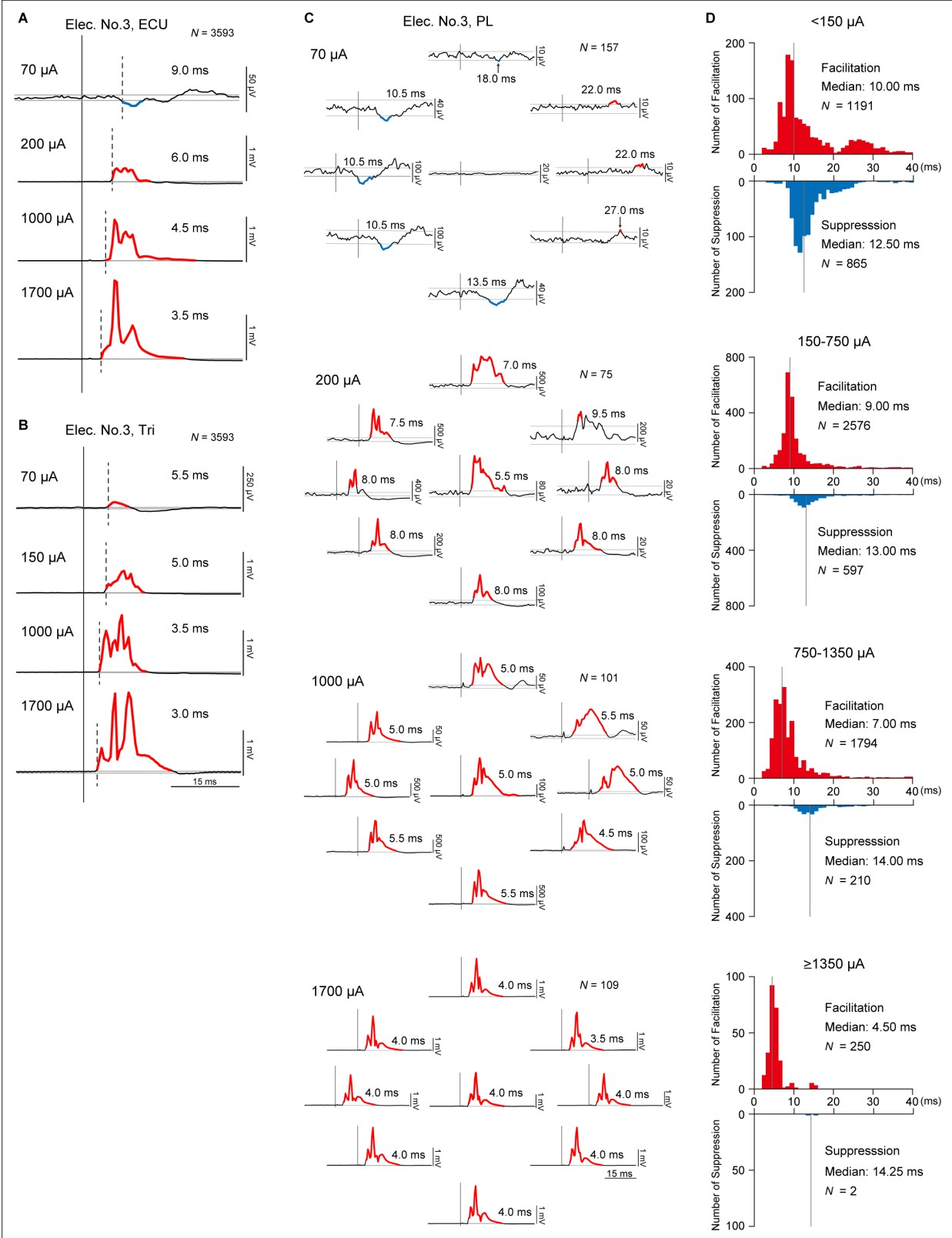

**Figure 8.** Latency of stimulus-induced muscle responses. (**A and B**) Post-stimulus effects (PStEs) and their onset latencies at four different current intensities during the whole period of the task. Dashed lines indicate the onset of responses. Red and blue lines indicate Facilitation effects and Suppression effects, respectively. *N* is the number of stimuli for averaged evoked muscle responses. (**A**) Examples of PStE changing from Facilitation to Suppression. (**B**) Examples of PStE for Facilitation. (**C**) Directional tuning of PStEs and their onset latencies in the palmaris longus (PL) muscle at four

*Figure 8 continued*

different current intensities through Elec. No. 3 during the hold period for each target. Red and blue lines show Facilitation and Suppression effects, respectively. *N* is the number of stimuli for averaged evoked muscle responses. (**D**) Distributions of onset latencies for Facilitation and Suppression effects at four different current intensities during the hold period for the eight peripheral targets. Gray vertical lines indicate the medians of onset latencies for Facilitation and Suppression effects.

a disynaptic link via inhibitory interneurons, while the facilitation effects are via excitatory monosynaptic premotoneuronal afferent fibers and/or facilitatory interneurons. The gain of such mono- and polysynaptic spinal reflexes depends on motoneuronal excitability, which is modulated by voluntary descending commands (*Capaday and Stein, 1986*; *Verrier, 1985*; *Zehr and Chua, 2000*). In line with these considerations, *Guiho et al., 2021* recently proposed a model of spinal circuitry driven by spinal electrical stimulation. In their model, the discharges of excitatory and inhibitory interneurons elicited by spinal stimulation are assumed to be integrated into motoneuron activity receiving corticospinal drives. The current study extends this model so that the voluntary descending commands are integrated with afferent inputs at spinal interneurons as well as motoneurons. Spinal stimulation at low and medium currents (<1350 µA) induced facilitation and/or suppression effects in multiple forelimb muscles (*Figures 3 and 6*). Magnitudes of these effects were proportional to the level of background EMG (*Figure 5*), which depends on the amount of descending commands and suggests that descending commands amplified the functions of intraspinal neural elements such as excitatory and inhibitory synaptic connections to motoneurons. This leads to the correspondence between PDs of stimulus-induced muscle responses and background EMG (*Figure 3F*, top and medium panels) and positive correlations between the magnitude of stimulus-induced muscle responses and background EMGs (*Figure 6E*, top and medium panels). The magnitude of stimulus-induced muscle responses would depend on excitability and number of spinal motoneurons and interneurons in the subliminal fringe.

On the other hand, higher currents (≥1350 µA) induced a large magnitude of facilitation effects in most muscles (*Figure 3C and D*) and showed the disappearance of Spinal PD (*Figure 3A* bottom panel) or no correlation between the magnitudes of background EMG and stimulus-induced muscle responses (*Figure 6E*, bottom-left panel). These results indicate that current spread to the ventral aspect of the spinal cord leads to direct activation of motor axons, as well as recruitment of ascending pathways in the dorsal columns and dorsal rootlets. Previous studies reported that the shortest latency of muscle responses in the forearm by intraspinal microstimulation in the cervical enlargement was 2.8 ms due to the direct excitation of spinal motoneurons axons (*Perlmutter et al., 1998*; *Takei and Seki, 2010*). Our results showed that the latency of stimulus-induced muscle responses was shorter at higher currents, and the shortest latency was 2.5 ms (*Figure 8D*) which corresponded with the results in the previous reports. Thus, subdural stimulation at higher currents in present study most likely resulted in the direct activation of motor axons. Also, higher currents evoked stereotypical torque responses in the ulnar-flexion direction irrespective of the presence/absence or the direction of voluntary torque production (*Figure 7D*). This result might be due to the number and volume of wrist flexor and ulnar muscles being greater than the antagonist muscles so that the evoked torques were induced in the ulnar-flexion direction. During low descending commands with lower background EMG, higher currents induced larger facilitatory muscle responses (*Figure 6D*, left), indicating that many subthreshold neural elements, including both neurons within the subliminal fringe and deeper membrane potentials, were activated by higher currents. In contrast, during higher descending commands with higher background EMG, many motoneurons and motor axons are under the refractory period. Thus, higher currents could activate the few remaining subthreshold motoneurons and motor axons, causing smaller facilitatory muscle responses compared with lower descending commands (*Figure 6D*, right). Accordingly, during a higher level of descending commands at higher currents, Spinal PD at higher currents became opposite of the PD of background EMGs (*Figure 3F*, bottom-left panel), and stimulus-induced muscle responses became independent or showed a negative correlation with background EMGs (*Figure 6D and E*, bottom-left panel).

As shown in our results of onset latency (*Figure 8D*), facilitation effects were observed at as weak intensity as suppression effects. However, results showed suppressed voluntary torques at lower currents (*Figure 7B*), indicating stronger suppression effects via inhibitory interneurons. The neural mechanisms underlying the suppressed voluntary torques are unclear, but afferent inputs mediated by

Ia presynaptic inhibition to motoneurons or autogenic inhibition to agonist motoneurons via inhibitory interneurons might be possible mechanisms for suppression effects at lower currents. The inhibitory influence mediated by presynaptic mechanisms (*Eccles et al., 1961*; *Eccles et al., 1962*; *Rudomin and Schmidt, 1999*) is known to act on afferents but not corticospinal inputs to motoneurons (*Jackson et al., 2006*). In addition, during muscular contraction, autogenic inhibition to agonist motoneurons via inhibitory interneurons is driven by Ib afferents (*Houk, 1979*; *Lundberg and Malmgren, 1988*). Our results suggest that lower currents predominantly result in these effects on the agonist motoneurons and suppress voluntary torque. The inter-muscular relationship characterized by the PDs of background EMGs in the wrist muscles (*Figure 7—figure supplement 1*) demonstrate that the monkeys consistently engaged each muscle as agonist and that antagonistic voluntary contractions were rare irrespective of stimulus currents (see polar plots of background EMGs of *Figure 7—figure supplement 1A–C*). This result indicates that the presumed different activation in the spinal excitatory and inhibitory interneurons at different current intensity is not supported by the change of wrist torque production strategy.

## Voluntary command tunes stimulus-induced muscle responses

Directional tuning of neural activity during voluntary movements is observed in each hierarchical neural element from the cerebral cortex to muscles. Neurons in motor-related areas of the cerebral cortex exhibit directional tuning during motor execution (*Caminiti et al., 1990*; *Cisek et al., 2003*; *Crammond and Kalaska, 1996*; *Fu et al., 1993*; *Georgopoulos et al., 1982*; *Georgopoulos et al., 1986*; *Schwartz et al., 1988*; *Kakei et al., 1999*; *Kakei et al., 2001*; *Sergio and Kalaska, 1997*). Activity of spinal interneurons (*Fetz et al., 2002*), muscles (*Buchanan et al., 1986*; *Hoffman and Strick, 1999*; *Kato et al., 2016*), and peripheral afferents (*Jones et al., 2001*) also shows directional tuning. Motor-evoked potentials induced by transcranial magnetic stimulation, which reflects the excitability of corticospinal tract and spinal motoneurons, are tuned by the movement direction (*Kadota et al., 2014*). In our results, the magnitude of induced muscle responses for both facilitation and suppression effects showed directional tuning (*Figure 2*) and correlated positively with the level of background muscle activity at lower currents (*Figure 6*). Furthermore, the PD of induced muscle responses was identical to the PD of background EMG (*Figure 2D*). Therefore, the voluntary commands for torque direction are associated tightly with the commands for the level of muscle activation, indicating that voluntary commands for movement directions determine the excitability of spinal motoneurons of each muscle. Compared with the hold period for the center target, the stimulus-induced muscle responses and torques at low to medium currents were generally more pronounced during the hold period for the peripheral targets (*Figure 2A–C*, *Figure 7B and C*, and *Figure 7—figure supplement 1*), indicating that the descending commands augmented activation in the spinal motoneurons and interneurons driven by spinal stimulation. Interestingly, at medium currents, the stimulus-induced facilitatory responses were sometimes smaller when the responses were recorded in the antagonistic muscles against the wrist torque direction regardless of the background EMG activity (*Figure 2A* and *Figure 7—figure supplement 1B*), suggesting that spinal reciprocal inhibitory function was evolved by the descending commands (*Meunier and Pierrot-Deseilligny, 1998*). Together, our findings indicate that voluntary commands amplify the functions of spinal circuits, including excitatory and inhibitory synaptic connections to motoneurons activated by spinal stimulation.

## Spinal stimulation activates divergent pathways

Spinal stimulations at both rostral and caudal sites induced facilitation and suppression effects to extensive upper-limb muscles (*Figure 4*). In addition to divergent innervations from afferent fibers at a distant stimulation site (*Brown et al., 1978*; *Brown and Fyffe, 1979*; *Brown and Fyffe, 1978*; *Ishizuka et al., 1979*), muscle responses in the caudally innervated muscles from rostral sites are thought to be generated via descending pathways in the dorsolateral funiculus, such as corticospinal and rubrospinal pathways, as well as descending propriospinal tracts. Muscle responses in rostrally innervated muscles from caudal sites are produced via ascending pathways, such as the spinocerebellar pathway, dorsal column-medial lemniscus pathway, and ascending propriospinal tracts. However, muscles innervated by motoneurons located near the stimulation site produced larger magnitudes of muscle responses (*Figure 4—figure supplement 1*, *Figure 4E*, left, and *Figure 4F*), indicating that the motor nucleus

for each muscle is innervated dominantly by adjacent afferent fibers rather than distant ones (**Brown et al., 1978**).

Stimulations at most stimulus sites showed that the PDs for facilitation and suppression effects were similar to those for background EMGs (**Figure 4G and H**). The correspondence in the PDs between the stimulus-induced muscle response and background EMGs indicates that changes in the amount of voluntary descending commands principally account for the torque direction-dependent modulation in the evoked muscle response. Such change in the evoked muscle response suggests that spinal stimulation produced trans-synaptic inputs to motoneurons that can be spatiotemporally summated by voluntary descending inputs. However, an exception was observed in some cases of rostrally innervated muscles that showed facilitation effects. The Spinal PDs for facilitation in the rostrally innervated muscles from caudal sites were opposite to those for background EMGs (**Figure 4G**, bottom-left panel). The magnitude of these responses was quite small (**Figure 4E**, left panel), but this feature of responses was similar to the response at higher current (**Figure 3F**, lower panel). These results suggest that some motoneurons of rostrally innervated muscles may not receive excitatory ascending inputs from afferents of the caudal part of the spinal site. Although there is a considerable distance between them, current targeting to the caudal site might spread to ventral roots of rostrally innervated muscles.

## Implications for clinical application

In general, an advantage of spinal stimulation is that a single electrode produces facilitation and suppression effects on synergistic muscle groups in multi-joints (**Kato et al., 2020**; **Nishimura et al., 2013**), which is different from neuromuscular electrical stimulation (NMES). Since NMES activates the motor end plates or muscle fibers directly, muscular contraction is accomplished with an inverted recruitment order that large diameter muscle fibers are preferentially activated (**McNeal, 1976**), leading to rapid fatigue (**Prochazka, 1993**). In contrast, spinal stimulation at an appropriate current recruited motoneurons trans-synaptically (**Figures 3 and 6**) via afferent fibers, so that motoneurons are activated in a natural recruitment order, which, in turn, reduces fatigue (**Bamford et al., 2005**).

Since our study aimed to capture fundamental characteristics of descending commands on motor outputs to spinal stimulation, we used single pulses for spinal stimulation and investigated their muscle responses during voluntary motor task, instead of high frequency stimulation which has been used for therapeutic spinal stimulation. Using single pulse stimulation, we were able to characterize suppressive effects on muscle responses and joint torque that were obtained when motoneurons were voluntarily preactivated. An important finding demonstrated in the present study is that the current of 150–1350 µA in the range where excitation and inhibition coordinated induces appropriate effects to enhance descending commands and functions of spinal circuits, thus, boost torque production in a direction corresponding with the direction of voluntary torque production (**Figure 7**). On the other hand, regardless of the directions of voluntary torque production, lower-current (<150 µA) stimulation suppressed torque, and higher-current (≥1350 µA) stimulation induced stereotypical torque (**Figure 7**), indicating that the induced torques at these current intensities interfere with voluntary commands. Thus, careful selection of current intensity is necessary to enhance voluntary torque production. We believe that our findings obtained in intact animals may be applicable to individuals with incomplete spinal cord injury or stroke in which the function of spinal circuits and descending pathways are preserved. In future studies, the current intensity that produces balanced excitation and inhibition for spinal stimulation should be used to compensate the weakened descending commands and restore impaired upper limb motor functions after damage to descending pathways.

Epidural stimulation of the spinal cord has been commonly used in the treatment for chronic pain (**Epstein and Palmieri, 2012**; **Compton et al., 2012**), and there is increasing interest in further applications for the restoration and/or rehabilitation of motor functions after damage in descending pathways (**Angeli et al., 2018**; **Gill et al., 2018**; **Harkema et al., 2011**; **Wagner et al., 2018**; **Barra et al., 2022**; **Capogrosso et al., 2016**; **Courtine et al., 2009**; **van den Brand et al., 2012**; **Wenger et al., 2016**). Subdural electrodes are more invasive than epidural electrodes but have the advantage of selectively activating a specific group of spinal motoneurons (**Sharpe and Jackson, 2014**). Therefore, subdural stimulation may be beneficial to restore dexterous hand control which requires independent control of muscles and fingers. However, the effectiveness of subdural stimulation in controlling

dexterous hand movements and the long-term stability of motor output needs to be determined in future studies.

## Materials and methods

The experiments were performed using two male Japanese macaque monkeys (*Macaca fuscata*; monkey H, weight 8.1 kg; monkey W, weight 4.9 kg). All experimental procedures were performed in accordance with the guidelines for the Ministry of Education, Culture, Sports, Science, and Technology (MEXT) of Japan and the Care and Use of Nonhuman Primates in Neuroscience Research (Japan Neuroscience Society) and were approved by the Institutional Animal Care and Use Committee of the Tokyo Metropolitan Institute of Medical Science (18035, 19050, 20–053, 21–048). Throughout the experiments, the monkeys were housed in individual cages at an ambient temperature of 23–26°C and a 12 hr on/off light cycle. The animals were fed regularly with diet pellets and had free access to water. They were monitored closely, and animal welfare was assessed on a daily basis or, if necessary, several times a day.

### Surgery

All surgeries were performed under sterile conditions and general anesthesia, starting with a combination of intramuscular injections of ketamine (5 mg/kg) and xylazine (0.5 mg/kg), followed by intubation and isoflurane (1–2%) inhalation to maintain anesthesia throughout surgery. During surgery, vital signs were carefully monitored, including respiratory/circulatory parameters (respiratory rate, inspiratory carbon dioxide concentration, saturation of percutaneous oxygen, and heart rate) and body temperature. There was no evidence of tachycardia or tachypnea during surgical procedures and no major deviation in the heart or respiratory rate in response to noxious stimuli. The absence of reflexive movements to noxious stimuli and a corneal reflex was also used to verify the level of anesthesia. Ceftriaxone (20 mg/kg) and ketoprofen (2 mg/kg) were administered preoperatively and postoperatively.

### Surgery to place a subdural electrode array on the spinal cord

We chronically implanted a platinum subdural electrode array (Unique Medical Corporation, Tokyo, Japan; *Figure 1A*) over the dorsal-lateral aspect of the cervical enlargement on right side (*Figure 1B*) corresponding to the hand performing the task. A subdural electrode array with seven channels was implanted in monkey H over the rostral C6 to caudal T1 region. A subdural electrode array with six channels was implanted in monkey W over the caudal C6 to rostral T2 region. The electrodes had a diameter of 1 mm and a center-to-center inter-electrode distance of 3 mm (*Figure 1A*). A silver plate (3×2 mm) placed on spinal vertebra was used as a reference electrode. In both monkeys, laminectomy was performed on the C7 vertebra, and the lamina and dorsal spinous process of C7 were removed. An incision was made in the dura mater under the C7 vertebra. In monkey W, laminectomy was also performed on the C4 vertebra, and an incision was made in the dura mater under the C4 vertebra. The subdural electrode array was slid into the subdural space from the caudal incision site on C7 vertebra level and placed over the dorsal-lateral aspect of the C6-T2 spinal segments on the right side (*Figure 1B*). The electrode array was bonded with cyanoacrylate glue to the spinal surfaces at each laminectomy point. The wires from the electrodes were routed into a silicone tube, which was glued with dental acrylic to bone screws placed in T1 spinal process, and routed toward the monkey's head to a connector. The laminectomy was covered with gelatin, and a reference electrode was inserted into the space between the dorsal cervical vertebrae and back muscles. The skin and back muscle incisions were closed with silk and nylon sutures, respectively.

### Surgery for EMG recording

For EMG, multi-stranded stainless-steel wires were surgically implanted in 16 arm and hand muscles on the right side that were identified by anatomical features and evoked movements elicited by trains of low-intensity stimulation to the muscles. Bipolar wires (Cooner Wire, Chatsworth, CA, USA) were sutured into each muscle, and the wires were routed subcutaneously to connectors (FTSH-118–04 L-D, Samtec, New Albany, IN, USA) that were anchored to the skull. In both monkeys, wires were implanted in the following 16 muscles: three elbow muscles (BB, BR, and Triceps), six wrist muscles (PT, FCR,

PL, FCU, ECU, and ECR), five digit muscles (FDS, FDP, EDC, ED4, 5, and APL), and two intrinsic hand muscles (ADP and ADM).

## Behavioral task

Prior to the surgeries, the monkeys were trained to perform an isometric, 2D, eight-target wrist torque tracking task. The monkeys controlled the 2D position of a cursor on a video monitor with four-directional wrist torques: flexion, extension, ulnar flexion, and radial flexion. The cursor was adjusted to the center of eight peripheral targets when the wrist torques were neutral. When the cursor stayed on the center position for 0.8 s, one of the eight cursors appeared as a go-cue instruction. Then, the monkey was required to maintain torque within the target for 0.7–0.8 s to receive a juice reward (*Figure 1C and D*). Each of eight targets was presented in randomized order. Each experiment consisted of 63–1004 successful trials (Table 2 in *Supplementary file 1*).

## Stimulus protocols

Spinal stimulation was administered while the monkeys performed the isometric, eight-target wrist torque tracking task or the monkeys were anesthetized. During the tracking task in a monkey chair, stimulation consisting of constant-current, biphasic square-wave pulses of 0.2 ms with an inter-stimulus interval of 197 ms was applied continuously using a single electrode on the spinal cord (*Figure 1B–D*).

In the experiments under anesthesia, the monkeys received intramuscular injections of ketamine (5 mg/kg). Then, monkey H was seated in a monkey chair with its head fixed in a frame attached to the chair, and monkey W was laid in lateral position on a table. Spinal stimuli consisting of three constant-current, biphasic square-wave pulses of 333 Hz with 0.2 ms duration were delivered through a single electrode on the spinal cord. Each stimulus train was delivered with an interval of 1000 ms. The evoked movements and muscle twitches were detected by visual inspection and further monitored by direct muscle palpation. The movement threshold was defined as the minimum current at which the evoked muscle twitch was observed by visual inspection. Additional doses of ketamine were given as needed to eliminate spontaneous movements during the recording sessions.

## Data collection

During the experiments, the trigger pulses of stimulation, EMGs recorded from the implanted wires into muscles, and task parameters, such as target positions, timing trial events, and wrist torques (torque X, flexion-extension; torque Y, radial-ulnar), were recorded simultaneously using a Cerebus multichannel data acquisition system (Blackrock Microsystems, Salt Lake City, UT, USA) at a sampling rate of 2 kHz. The EMGs were bandpass filtered at 5–1000 Hz for offline analysis.

## Data analysis
### StTA of rectified EMGs

Muscle responses were investigated using the StTA of rectified EMGs at each current intensity and stimulus site (*Cheney et al., 1985b*; *Cheney and Fetz, 1985a*). The StTA of rectified EMGs depicts both facilitative and suppressive effects induced by stimulation. The averages of rectified EMG data were compiled over a 100 ms period (30 ms before the trigger to 70 ms after). To avoid contamination by stimulus artifacts, EMG signals for 0–2 ms after stimulation were excluded for analyses. Mean baseline activity and SD were measured from EMGs in the period from 30 to 10 ms preceding the stimulus trigger pulse. The significant stimulus-evoked facilitative (Facilitation) or suppressive (Suppression) effects were detected as sustained features (total duration of ≥1 ms) above or below 3 SD from the mean baseline, respectively (*Figure 1E*). Onset latency was defined as the beginning of the Facilitation or Suppression effects. The magnitude of the PStEs was quantified as the area above or below 3 SD from the mean baseline (red and blue hatched areas on *Figure 1E*). PStEs typically showed an intermixture of Facilitation and Suppression at different latency in a single muscle (e.g. Facilitation followed by Suppression [first row in *Figure 1E*] or Suppression followed by Facilitation [fourth row in *Figure 1E*]). The present analyses focused on the first PStE, which appeared in shorter latency. Therefore, PStE induced by spinal stimulation showed either Facilitation or Suppression effect for the result of each StTA. When both Facilitation and Suppression were obtained from a single muscle (i.e. PStEs for Facilitation and Suppression changed by directions of voluntary torques), a dominant effect, which is determined by the comparison between the sums of each PStE for the eight target locations, was

indicated with a plus symbol. PStEs were collected from a total of 1008 muscular conditions (Table 1 in *Supplementary file 1*). A single muscular condition was defined as the result of PStEs (*Figures 2–6*) or torques (*Figure 7*) observed from a single muscle in an experiment. For example, when PStEs for the eight-target locations (*Figures 2–4*) or PStEs for five different levels of background EMG (*Figures 5 and 6*) from a single muscle were produced, the result of 8-PStE or 5-PStE corresponded to one muscular condition. Background EMG was defined as muscle activity in the period from 30 to 10 ms preceding the stimulus trigger pulse.

### Directional tuning of the induced muscle responses and the background muscle activities

We examined whether PStE (Facilitation or Suppression) and background EMG show directional tuning during the task (*Figures 2–4*). To test the presence of significant directional modulation, the background EMG and PStE data were shuffled separately with respect to the torque directions. The vector of PD was then calculated from this shuffled data. This process was repeated 1000 times, and the distribution of the angle of the PD vector was sorted by rank. The significance of PD angles for the background EMG and PStE (Spinal PD) was determined by computing the 95% bootstrap confidence interval (CI) of the sorted angular distribution and comparing with the actual angular data. A PD angle was significant if the actual angle fell outside the 95% CI of the distribution of bootstrapped angular data ($p < 0.05$). To investigate absolute differences between the PD of background EMG and the Spinal PD, the Spinal PD angle was normalized by subtracting the PD of background EMG (Normalized Spinal PD).

### Relationship between the magnitude of PStEs and the background EMGs

To investigate the relationship between the magnitudes of the background EMGs and PStEs (*Figures 5 and 6*), we classified individual peristimulus muscle activity into five different levels based on the magnitudes of background EMGs. Then, each level of muscle activity was separately averaged to examine the PStEs. Two-sided Pearson's correlation coefficients were computed between the magnitudes of the background EMGs and PStEs.

### StTA of wrist torques

In addition to the muscle responses, the induced wrist torques were investigated using the StTA. The averages of torque data were compiled over a 180 ms period (30 ms before the trigger to 150 ms after). To detrend the baseline from raw StTA of torque, we fitted the baseline trends by a polynomial of appropriate order (1–4) and subtracted this from the raw StTA of torque. Mean baseline and SD were measured from the subtracted torque data in the period from 30 to 10 ms preceding the stimulus trigger pulse. The significant induced torques (Evoked Torque) were detected when the torque in either the flexion-extension or radial-ulnar direction or both were 10 SD above or below the mean baseline (*Figure 1E*). The magnitude and direction of Evoked Torque were measured as the distance and angles from average of the baseline torque trajectory to the farthest point of the trajectory, respectively.

### Directional tuning of wrist torques

To investigate absolute differences between the direction of Evoked Torque and voluntary torque production, the Evoked Torque angle was normalized by subtracting the direction of voluntary torque production (Normalized Torque).

### Statistical analysis for population data

To determine the differences in magnitudes of Facilitation or Suppression effects on muscle activity by current intensity (*Figure 3D*) or stimulus site (*Figure 4E and F*), we computed the sum of Facilitation or Suppression during the hold period of a wrist torque for the eight target locations. Then, we performed one-way factorial ANOVA with Tukey-Kramer correction for post hoc multiple comparisons (*Figure 3D*) or two-sided unpaired t-test (*Figure 4E and F*). Likewise, the differences in magnitudes of Evoked Torque by current intensity were examined by ANOVA with Tukey-Kramer correction for post hoc multiple comparisons (*Figure 7E*).

In analysis of population data of PDs, the PDs of background EMG were categorized into the 'background EMG PD' for Facilitation or Suppression by whether the muscular condition induced Spinal PD for Facilitation and/or Suppression (*Figures 2 and 4*). If the muscular condition induced both Facilitation and Suppression (i.e. PStEs changed by movement directions), the PD of background EMG was included into both the background EMG PDs of Facilitation and Suppression. To test whether distributions of Spinal PD, PD of background EMG, and directions of Evoked Torque showed any directional preference, we used Rayleigh test for uniformity of the population (*Figures 2–4* and *Figure 7*). V-test was used to determine whether the observed angles cluster around the predicted angles, and the distribution are significantly different from randomness (*Batschelet, 1981*, *Figure 2D*, *Figure 3E, F*, *Figure 4G, H* and *Figure 7F*). We hypothesized that distributions for Spinal PD, background EMG PD, and the PD for Evoked Torque are tuned to the circular medians. Also, we assumed that distributions for Normalized Spinal PD and the PD for Normalized Torque tune around 0° in the case of trans-synaptic recruitment of motoneurons or 180° in the case of direct activation of motoneurons due to stimulation at high currents.

## Acknowledgements

We thank N Hashimoto, S Nagai, and M Hashimoto for technical help and E Wakatsuki for assistance with the illustration. This work was supported by grants from JSPS KAKENHI (Grant Number 18H05287, 18H04038, 20H05489, and 20H05713 to Y.N., and JP18J11771 and 20K19377 to M.K.) and by JST (Moonshot R&D - MILLENNIA Program) Grant Number (JPMJMS2012) to Y.N.

## Additional information

### Funding

| Funder | Grant reference number | Author |
| --- | --- | --- |
| Japan Society for the Promotion of Science | KAKENHI 18H05287 | Yukio Nishimura |
| Japan Society for the Promotion of Science | KAKENHI 18H04038 | Yukio Nishimura |
| Japan Society for the Promotion of Science | KAKENHI 20H05489 | Yukio Nishimura |
| Japan Society for the Promotion of Science | KAKENHI 20H05713 | Yukio Nishimura |
| Japan Society for the Promotion of Science | KAKENHI JP18J11771 | Miki Kaneshige |
| Japan Society for the Promotion of Science | KAKENHI 20K19377 | Miki Kaneshige |
| Moonshot Research and Development Program | JPMJMS2012 | Yukio Nishimura |

The funders had no role in study design, data collection and interpretation, or the decision to submit the work for publication.

### Author contributions

Miki Kaneshige, Conceptualization, Data curation, Formal analysis, Funding acquisition, Investigation, Visualization, Methodology, Writing – original draft, Writing – review and editing; Kei Obara, Conceptualization, Data curation, Formal analysis, Investigation, Methodology; Michiaki Suzuki, Methodology; Toshiki Tazoe, Writing – review and editing; Yukio Nishimura, Conceptualization, Supervision, Funding acquisition, Visualization, Methodology, Project administration, Writing – review and editing

### Author ORCIDs

Miki Kaneshige ⬤ http://orcid.org/0000-0002-0808-5454
Michiaki Suzuki ⬤ http://orcid.org/0000-0003-0725-0515
Yukio Nishimura ⬤ http://orcid.org/0000-0003-3479-2483

### Ethics

All experimental procedures were performed in accordance with the guidelines for the Ministry of Education, Culture, Sports, Science, and Technology (MEXT) of Japan and the Care and Use of Nonhuman Primates in Neuroscience Research (Japan Neuroscience Society) and were approved by the Institutional Animal Care and Use Committee of the Tokyo Metropolitan Institute of Medical Science (18035, 19050, 20-053, 21-048).

### Decision letter and Author response

Decision letter https://doi.org/10.7554/eLife.78346.sa1
Author response https://doi.org/10.7554/eLife.78346.sa2

## Additional files

### Supplementary files

- Transparent reporting form
- Supplementary file 1. Summary of collected data.

### Data availability

Figure 2 - Source Data 1, Figure 3 - Source Data 1, Figure 3 - Source Data 2, Figure 3 - Source Data 3, Figure 4 - Source Data 1, Figure 4 - Source Data 2, Figure 4 - Source Data 3, Figure 4 - Source Data 4, Figure 7 - Source Data 1 and Figure 7 - Source Data 2 contain the numerical data used to generate the figures.

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
