## [Editor Report]

This study will be of interest to anyone wishing to develop neurotechnologies for restoring motor control following injury. The results convincingly show that spinal stimulation could facilitate or suppress voluntary muscle engagement and joint movement, depending on both the voluntarily evoked activity and the stimulation parameters. This finding is important, as it provides new opportunities for improving stimulation guided neurorehabilitation, particularly in cases of partial lesions.

---

## [Decision Letter]

**Decision letter after peer review:**

Thank you for submitting your article "Tuning of motor outputs produced by spinal stimulation during voluntary control of torque directions in monkeys" for consideration by *eLife*. Your article has been reviewed by 3 peer reviewers, one of whom is a member of our Board of Reviewing Editors, and the evaluation has been overseen by Tamar Makin as the Senior Editor. The following individual involved in the review of your submission has agreed to reveal their identity: Santosh Chandrasekaran (Reviewer #2).

Essential revisions:

All reviewers highly encourage the authors to include latency data (excitatory, inhibitory, and at high current intensities), as this is essential for the interpretation of the results. We hope this should be straightforward to include. The reviewers also all agreed that it would be useful to include examples and discussion of evoked responses during awake rest. If these data are available, they would substantially strengthen the results and interpretation of the manuscript.

In addition, there is a range of specific suggestions from the three reviewers to improve the clarity and readability of the manuscript. These are specific points and full reviews are listed below.

*Reviewer #1 (Recommendations for the authors):*

The paper is densely written but readable. The main conclusions are supported adequately.

Several general points can improve the presentation:

The possible differences between subdural and epidural stimulation regimes are not adequately discussed in the evaluation of the results in the broader context. Similarly, the discussion should provide more discussion of the meaning of the data here and limitations in the context of e.g., incomplete spinal cord injury (SCI) or discomplete SCI. In contexts after injury with variable descending modulatory controls, will the intact mechanisms here transfer? For example, could the stimulation after SCI ever worsen the quality of voluntary control through residual surviving pathways?

It would be very useful to supply an analysis of the distributions of latency of the inhibitory effects and excitatory effects: (1) all inhibitory or excitatory together, or (2) broken out by response type (E, I, E/I +/-). Also please discuss does latency vary much with stimulus strength? These statistics would add further information to the data here and help readers interpret some aspects.

The term 'conditions' is used repeatedly but is not well-defined. This term should be defined and used in a more precise way. (see lines 125, 138).

The term Spinal stim refers I think to the anesthetized experiment but should be better defined and qualified.

It would be useful in the section starting on Line 189 to create a plot of angular deviation of directional tuning from say a direction at 200uA and how this deviation and its variance changes with stimulation intensity. Presumably, there is a trade-off between the deviation and strength of effect which would in clinical conditions be important.

The term optimization is used a lot but this is not really correct. There is no optimization. Better would be 'range of balanced excitation and inhibition' or some such terminology.

A 'V-test' is mentioned at the end of methods but never described or citation provided and the term is highly ambiguous and needs clarifying.

Specific Comments:

Abstract:

The main result seems more consistent with stimulus amplifying the voluntary control, or else voluntary control both amplifying and shaping the stimulation effect. The sentence:

"Here, we show that descending commands amplify stimulus-evoked joint torque and the function of intraspinal elements." Does not capture completely accurately my understanding of the results.

Lines 37-42: It is important to clearly distinguish ISMS and supraspinal stimulation literature here, since the access and recruitment selectivity may be very different.

Bizzi, Mussa-Ivaldi and Giszter, Science 1991, Giszter Mussa-Ivaldi and Bizzi, J Neurosci 1993 and Mussa-Ivaldi, Giszter and Bizzi PNAS 1994 should likely be added to Loeb 1993 for spinal. Also Lemay and Grill, J Neurophys 2004l,

Line 44-45 In relation to the conclusion in this sentence, it is likely important to note that both Mussa-Ivaldi et al. 1994 and Lemay and Grill 2004 showed winner-take-all effects of stimulation in some instances, likely a strong indicator of inhibition mechanisms, although at points well above the drive to motor pools.

Line 52 -53 The Sharma and Shah J Physiol 2021 cervical spinal stimulation study in rats should be cited here or lines 60-61.

Line 63 the differences possible in subdural versus epidural stimulation regimes needs to be elaborated here or in discussion.

Line 76 rootlets (plural).

Line 88-89 Please also supply the current level here.

Line 178 Legend Part D:

"Distributions of the PDs for PStEs (Spinal PD, top panels), PDs for background

EMGs that induced PStEs (Background EMG PD, middle panels) and normalized PDs for PStEs (Normalized Spinal PD, bottom panels)." This is not very clear and should be unpacked.

Line 275 two different rostral and caudal sites?? Clarify.

Line 307 Is it fair to say the effect is tuned or biased to the adjacent motor pools here?

Figure 5: Part D. Readers need a key to the hatching. This is never explained adequately here. Also, have you explored displaying this data after a Fisher-Z transform of the correlations?

Figure 6 – same hatching issues – no description provided.

Line 505-506 There should be citations here also of the work of Drs Takei and Seki.

Line 546 Should potential current spread recruiting dorsal columns be discussed here?

Line 569 The Ib recruitment explanation here seems naive. Couldn't various inhibitory pathways be more subliminal and 'in reserve' in normal voluntary control until stronger activity excites them and more Ia (and Ib) inputs combine?

Line 587 "Stimulations at most stimulus sites showed that the PDs for facilitation and suppression effects were similar to those for background EMGs (Figure 4G and H), indicating the trans-synaptic recruitment of motoneurons" … Please explain this better, unpack it for the reader.

Line 609 There is no real optimization here, just a description of a band where both excitation and inhibition are coordinated.

Line 780 – uncited reference to a 'V-test'.

*Reviewer #2 (Recommendations for the authors):*

1. In my opinion, it would benefit the readability of the manuscript if the authors change the statement discussed in my 1st comment to "interaction of spinal stimulation with voluntarily evoked EMG and torque outputs" throughout the manuscript.

2. Why was the spinal stimulation ISI 197 ms, i.e., at approx. 5 Hz? Usually, therapeutic epidural stimulation is delivered at 50 Hz.

3. The authors do not state how many trials were performed per current level? Were these results tested for stability across days? Given that voluntarily evoked activity could vary based on factors such as motivation, how does that affect the efficacy of spinal stimulation?

4. The authors discuss the results in relation to polysynaptic inhibition and monosynaptic excitation that are present in the spinal circuits. I would highly encourage the authors to cite Guiho et al., JNE 2021 which proposes a potential novel model of spinal circuitry, and include it in the discussion.

*Reviewer #3 (Recommendations for the authors):*

Overall, the manuscript is clear and compelling. Some modifications to the figures and text could help to improve clarity:

1. In most figures, the PD vector uses the same color and line width as the rest of the polar plot. It would be helpful to highlight the PD vector by making the line thicker or a slightly different color.

2. It is not currently clear how different levels of background EMG activity were achieved. Were the monkeys trained to vary this activity level or were these differences analyzed post hoc based on the activity they happened to produce during various trials?

3. In Figure 5, I think the plots show increasing background activity moving from left to right, but this isn't currently labeled in the figure.

4. The descriptions of Figure 7 are confusing (e.g. center of peripheral panels, outer peripheral panels). It might be helpful to add an additional panel to the figure that shows the structure of each of the other panels without any data but including labels to help clarify what the different parts of each panel are showing.

5. It would be helpful to redefine the abbreviations for muscles in the methods section (i.e. at lines 670-672).

---

## [Author Response]

Essential revisions:All reviewers highly encourage the authors to include latency data (excitatory, inhibitory, and at high current intensities), as this is essential for the interpretation of the results. We hope this should be straightforward to include. The reviewers also all agreed that it would be useful to include examples and discussion of evoked responses during awake rest. If these data are available, they would substantially strengthen the results and interpretation of the manuscript.

We analyzed our data of stimulus evoked responses further to provide the requested information about the response latency and the evoked responses during awake rest. We added the results of latency as Figure 8 and related descriptions in the Results and Discussion sections as described below. We also added examples of evoked responses during awake rest in center panels of Figures 2A-C, 5A-C, 6A-D and 7B-D, and the relating discussion as described below. Please confirm the details of their revisions in the individual responses to the reviewers’ specific comments. We believe that these additional data and discussions have strengthened the results and our manuscript.

Reviewer #1 (Recommendations for the authors):The paper is densely written but readable. The main conclusions are supported adequately.Several general points can improve the presentation:The possible differences between subdural and epidural stimulation regimes are not adequately discussed in the evaluation of the results in the broader context. Similarly, the discussion should provide more discussion of the meaning of the data here and limitations in the context of e.g., incomplete spinal cord injury (SCI) or discomplete SCI. In contexts after injury with variable descending modulatory controls, will the intact mechanisms here transfer? For example, could the stimulation after SCI ever worsen the quality of voluntary control through residual surviving pathways?

We added discussion about the differences between subdural and epidural stimulation (lines 742-752) and the implication of our data and the limitations of spinal stimulation in patients with incomplete SCI in the Discussion section as follows (lines 722-741):

Lines 742-752

“Epidural stimulation of the spinal cord has been commonly used in the treatment for chronic pain (Epstein and Palmieri 2012, Compton et al. 2012), and there is increasing interest in further applications for the restoration and/or rehabilitation of motor functions after damage in descending pathways (Angeli et al., 2018; Gill et al., 2018; Harkema et al., 2011; Wagner et al., 2018; Barra et al., 2022; Capogrosso et al., 2016; Courtine et al., 2009; Van Den Brand et al., 2012; Wenger et al., 2016). Subdural electrodes are more invasive than epidural electrodes, but have the advantage of selectively activating a specific group of spinal motoneurons (Sharpe and Jackson, 2014). Therefore, subdural stimulation may be beneficial to restore dexterous hand control which requires independent control of muscles and fingers. However, the effectiveness of subdural stimulation in controlling dexterous hand movements and the long-term stability of motor output need to be determined in future studies.”

Lines 722-741

“Since our study aimed to capture fundamental characteristics of descending commands on motor outputs to spinal stimulation, we used single pulses for spinal stimulation and investigated their muscle responses during voluntary motor task, instead of high frequency stimulation which has been used for therapeutic spinal stimulation. Using single pulse stimulation, we were able to characterize suppressive effects on muscle responses and joint torque that were obtained when motoneurons were voluntarily preactivated. An important finding demonstrated in the present study is that the current of 150-1350 μA in the range where excitation and inhibition coordinated induces appropriate effects to enhance descending commands and functions of spinal circuits, thus, boost torque production in a direction corresponding with the direction of voluntary torque production (Figure 7). On the other hand, regardless of the directions of voluntary torque production, lower-current (< 150 μA) stimulation suppressed torque and higher-current (≥ 1350 μA) stimulation induced stereotypical torque (Figure 7), indicating that the induced torques at these current intensities interfere with voluntary commands. Thus, careful selection of current intensity is necessary to enhance voluntary torque production. We believe that our findings obtained in intact animals may be applicable to individuals with incomplete SCI or stroke in which the function of spinal circuits and descending pathways are preserved. In future studies, the current intensity that produces balanced excitation and inhibition for spinal stimulation should be used to compensate the weakened descending commands and restore impaired upper limb motor functions after damage to descending pathways.”

It would be very useful to supply an analysis of the distributions of latency of the inhibitory effects and excitatory effects: (1) all inhibitory or excitatory together, or (2) broken out by response type (E, I, E/I +/-). Also please discuss does latency vary much with stimulus strength? These statistics would add further information to the data here and help readers interpret some aspects.

We added results and discussion regarding the distributions of latency of Facilitation and Suppression (lines 517-538, 540-551, 576-577, 610-616, 862-863).

Results

Lines 517-538

“Onset latency of evoked muscle responses altered by different current intensities

We found that stimulations at different current intensities induced different types of evoked muscle responses (Figure 3C), magnitudes of evoked muscle responses (Figure 3D), relations between muscle responses and background EMGs (Figure 6E), and directions and magnitudes of evoked torque (Figure 7F). However, it is unclear whether different current intensities of stimulations recruit different pathways. To answer this question, we examined the onset latencies of PStEs. Figure 8A and B exhibit typical examples showing effect of current intensity on onset latencies of PStEs. Regardless of whether the current intensity changed the type of PStEs (Figure 8A and B), the onset latency of the PStEs was found to shorten as the current increased.

We further examined effect of directions of voluntary torque and onset latencies of PStEs. Figure 8C shows PStEs in the PL muscle at four different current intensities during the hold periods for the 8-peripheral targets. In this example, stimulation at lower current (70 μA) clearly changed onset latencies of PStEs depending on directions of voluntary torque, while onset latencies of PStEs at higher currents (1000 and 1700 μA) were similar among different directions. Population data in Figure 8D shows the distributions of onset latencies of PStEs at four different current intensities during the hold period for the peripheral targets. The median onset latency of Facilitation effects was shorter than that of Suppression effects at all current intensities. In addition, the median onset latencies of Facilitation effects became shorter as stimulus currents increased, while those of Suppression effects were similar among the four stimulus currents. These current-dependent changes in onset latency indicate that the stimulus current affects recruited neural elements.”

Legend of Figure 8

Lines 540-551

“Figure 8. Latency of stimulus-induced muscle responses. (A, B) PStEs and their onset latencies at four different current intensities during the whole period of the task. Dashed lines indicate the onset of responses. Red and blue lines indicate Facilitation effects and Suppression effects, respectively. *N* is the number of stimuli for averaged evoked muscle responses. (A) Examples of PStE changing from Facilitation to Suppression. (B) Examples of PStE for Facilitation. (C) Directional tuning of PStEs and their onset latencies in the PL muscle at four different current intensities through Elec. No. 3 during the hold period for each target. Red and blue lines show Facilitation and Suppression effects, respectively. *N* is the number of stimuli for averaged evoked muscle responses. (D) Distributions of onset latencies for Facilitation and Suppression effects at four different current intensities during the hold period for the 8 peripheral targets. Gray vertical lines indicate the medians of onset latencies for Facilitation and Suppression effects.”

Discussion

Lines 576-577

“Onset latencies of evoked muscle responses were shorter with increasing current intensities (Figure 8A, B and D).”

Lines 610-616

“Previous studies reported that the shortest latency of muscle responses in the forearm by intraspinal microstimulation in the cervical enlargement was 2.8 ms due to the direct excitation of spinal motoneurons axons (Perlmutter et al., 1998; Takei and Seki, 2010). Our results showed that latency of stimulus-induced muscle responses was shorter at higher currents and the shortest latency was 2.5 ms (Figure 8D) which corresponded with the results in the previous reports. Thus, subdural stimulation at higher currents in present study most likely resulted in the direct activation of motor axons.”

Materials and methods.

Lines 862-863

“Onset latency was defined as the beginning of the Facilitation or Suppression effects.”

The term 'conditions' is used repeatedly but is not well-defined. This term should be defined and used in a more precise way. (see lines 125, 138).

We changed the term ‘conditions’ to the term ‘muscular conditions’, and added the definition in the Materials and methods section as follows:

Lines 873-878

“PStEs were collected from a total of 1008 muscular conditions (Table 1 in Supplementary File 1). A single muscular condition was defined as the result of PStEs (Figures 2-6) or torques (Figure 7) observed from a single muscle in an experiment. For example, when PStEs for the 8-target locations (Figures 2-4) or PStEs for five different levels of background EMG (Figures 5 and 6) from a single muscle were produced, the result of 8-PStE or 5-PStE corresponded to 1 muscular condition.”

The term Spinal stim refers I think to the anesthetized experiment but should be better defined and qualified.

We have not used the term Spinal stim in text, figures, or tables, but we first used the term “anesthesia” instead of “sedation”, and explained the stimulus parameters of “spinal stimulation under anesthesia” (lines 101-104) in more detail in the Results section as follows:

Lines 101-104

“To characterize spinal sites, we investigated evoked limb movements induced by spinal stimulation under anesthesia. Subdural spinal stimuli consisting of three constant-current, biphasic square-wave pulses of 333 Hz with 0.2 ms duration were delivered to the anesthetized monkeys through a single electrode on the spinal cord.”

It would be useful in the section starting on Line 189 to create a plot of angular deviation of directional tuning from say a direction at 200uA and how this deviation and its variance changes with stimulation intensity. Presumably, there is a trade-off between the deviation and strength of effect which would in clinical conditions be important.

Results of directional tuning for PStEs (Spinal PD) exhibited in Figures 3A and 3B have been included in population data of Figure 3E. Also, the differences between the Spinal PD and Background EMG PD have been indicated as Normalized Spinal PD in Figure 3F. The distributions of Normalized Spinal PD (Figure 3F) exhibited a decrease in correspondence between Spinal PD for Facilitation and Background EMG PD with increasing current intensities. To clarify the association between typical examples in Figure 3A and 3B and the population data in Figure 3E and 3F, we added an explanation in the Results section as follows:

Lines 229-233

“These results, in which low currents induced the Spinal PD similar to the PD of background EMG and large magnitudes of stimulus effects were induced in the direction of large magnitudes of background EMG, while high currents produced the Spinal PD opposite to the PD of background EMG, correspond to the typical examples of Figure 3A and 3B.”

The term optimization is used a lot but this is not really correct. There is no optimization. Better would be 'range of balanced excitation and inhibition' or some such terminology.

We replaced the term “optimized stimulus currents” by “stimulus currents in the range of balanced excitation and inhibition” or such explanations at lines 20-21, 566-567, 728-729 and 738-739.

A 'V-test' is mentioned at the end of methods but never described or citation provided and the term is highly ambiguous and needs clarifying.

We added a citation and explanation of the ‘V-test’ in the Materials and methods section as follows:

Lines 928-935

“V-test was used to determine whether the observed angles cluster around the predicted angles and the distribution are significantly different from randomness (Batschelet, 1981, Figure 2D, 3E, F, 4G, H, and 7F). We hypothesized that distributions for Spinal PD, Background EMG PD, and the PD for Evoked Torque are tuned to the circular medians. Also, we assumed that distributions for Normalized Spinal PD and the PD for Normalized Torque tune around 0 degrees in the case of trans-synaptic recruitment of motoneurons or 180 degrees in the case of direct activation of motoneurons due to stimulation at high currents.”

Added reference:

Batschelet E. 1981. Circular Statistics in Biology. London: Academic Press.

Specific Comments:Abstract:The main result seems more consistent with stimulus amplifying the voluntary control, or else voluntary control both amplifying and shaping the stimulation effect. The sentence:"Here, we show that descending commands amplify stimulus-evoked joint torque and the function of intraspinal elements." Does not capture completely accurately my understanding of the results.

We aimed to demonstrate how descending commands influence motor outputs driven by spinal stimulation. Therefore, we replaced the sentence as follows:

Lines 18-19

“Here, we show that descending commands amplify and shape the stimulus-induced muscle responses and torque outputs.”

Lines 37-42: It is important to clearly distinguish ISMS and supraspinal stimulation literature here, since the access and recruitment selectivity may be very different.Bizzi, Mussa-Ivaldi and Giszter, Science 1991, Giszter Mussa-Ivaldi and Bizzi, J Neurosci 1993 and Mussa-Ivaldi, Giszter and Bizzi PNAS 1994 should likely be added to Loeb 1993 for spinal. Also Lemay and Grill, J Neurophys 2004l,

We added the suggested citation as described below. Also, we distinguished the literature on ISMS and epidural spinal cord stimulation as follows:

Lines 36-44

“Motor outputs of spinal stimulation have been examined extensively and showed excitatory effects in anesthetized animals (epidural spinal cord stimulation: Greiner et al., 2021; intraspinal microstimulation: Moritz et al., 2007; Mushahwar et al., 2004; Zimmermann et al., 2011), spinalized animals (epidural spinal cord stimulation: Courtine et al., 2009; Van Den Brand et al., 2012; Capogrosso et al., 2016; Wenger et al., 2016; Barra et al., 2022; intraspinal microstimulation: Nishimura et al., 2013; Kasten et al., 2013; Mushahwar et al., 2004; Loeb et al., 1993; Tresch and Bizzi, 1999; Bizzi et al., 1991; Giszter et al., 1993; Mussa-Ivaldi et al., 1994), and humans (epidural spinal cord stimulation: Angeli et al., 2018; Gill et al., 2018; Harkema et al., 2011; Wagner et al., 2018).”

Added reference:

Bizzi E, Mussa-Ivaldi FA, Giszter S. 1991. Computations underlying the execution of movement: A biological perspective. Science 253:287–291. doi:10.1126/science.1857964

Giszter SF, Mussa-Ivaldi FA, Bizzi E. 1993. Convergent force fields organized in the frog’s spinal cord. J Neurosci 13:467–491. doi:10.1523/jneurosci.13-02-00467.1993

Mussa-Ivaldi FA, Giszter SF, Bizzi E. 1994. Linear combinations of primitives in vertebrate motor control. Proc Natl Acad Sci U S A 91:7534–7538. doi:10.1073/pnas.91.16.7534

Line 44-45 In relation to the conclusion in this sentence, it is likely important to note that both Mussa-Ivaldi et al. 1994 and Lemay and Grill 2004 showed winner-take-all effects of stimulation in some instances, likely a strong indicator of inhibition mechanisms, although at points well above the drive to motor pools.

We did not include these studies because they did not show direct evidence of contribution of inhibitory spinal interneurons to evoked movements.

Line 52 -53 The Sharma and Shah J Physiol 2021 cervical spinal stimulation study in rats should be cited here or lines 60-61.

We added the citation and revised the sentence as follows:

Lines 54-57

“Although a few studies have examined spinal stimulation in awake animals (Kato et al., 2020; Sharma and Shah, 2021; Barra et al., 2022), the modulation of muscle responses to spinal stimulation by descending commands has not been fully clarified.”

Added reference:

Sharma P, Shah PK. 2021. in vivo electrophysiological mechanisms underlying cervical epidural stimulation in adult rats. J Physiol 599:3121–3150. doi:10.1113/JP281146

Line 63 the differences possible in subdural versus epidural stimulation regimes needs to be elaborated here or in discussion.

We added a discussion of differences between subdural and epidural stimulation. Please refer the response to your first comment.

Line 76 rootlets (plural).

We corrected the term as “rootlets”.

Line 88-89 Please also supply the current level here.

We added information concerning current level as follows:

Lines 88-90

“One pulse of a biphasic square-wave with a duration of 0.2 ms and an interval of 197 ms was applied at stimulus currents of 110 µA through a single electrode during the task.”

Line 178 Legend Part D:"Distributions of the PDs for PStEs (Spinal PD, top panels), PDs for backgroundEMGs that induced PStEs (Background EMG PD, middle panels) and normalized PDs for PStEs (Normalized Spinal PD, bottom panels)." This is not very clear and should be unpacked.

We revised the sentence as follows:

Lines 182-187

“(D) Distributions of the Spinal, Background EMG, and Normalized Spinal PDs. Spinal PD (top panels) and Background EMG PD (middle panels) show the PDs calculated by the magnitudes of Facilitation or Suppression of PStEs and by the magnitudes of background EMG activity, respectively, during the hold period for the peripheral targets. Normalized Spinal PD (bottom panels) shows angles normalized by subtracting the Background EMG PD from the Spinal PD.”

Line 275 two different rostral and caudal sites?? Clarify.

Lines 280-281

We replaced “two different sites” by “two different rostral (Elec. No. 1) and caudal (Elec. No. 5) sites”.

Line 307 Is it fair to say the effect is tuned or biased to the adjacent motor pools here?

It might be possible that post-stimulus effects are tuned depending on the distance in the spinal levels from stimulus site to motor pools as different magnitude of stimulus-induced muscle response were observed. However, Spinal PDs corresponded to Background EMG PDs irrespective of the distance between stimulus sites to motor pools, indicating trans-synaptic excitation of motoneurons by activating a sufficient number of their inputs, such as afferent, propriospinal, and/or corticospinal axons. One exception was that the Spinal PD was opposite to the directions of background EMG PDs in rostral innervated muscles from caudal spinal site. We discussed this point in the text (lines 702-710).

Lines 702-710

“However, an exception was observed in some cases of rostrally-innervated muscles that showed facilitation effects. The Spinal PDs for facilitation in the rostrally-innervated muscles from caudal sites were opposite to those for background EMGs (Figure 4G, bottom-left panel). The magnitude of these responses was quite small (Figure 4E, left panel), but this feature of responses was similar to the response at higher current (Figure 3F, lower panel). These results suggest that some motoneurons of rostrally-innervated muscles may not receive excitatory ascending inputs from afferents of the caudal part of the spinal site. Although there is a considerable distance between them, current targeting to the caudal site might spread to ventral roots of rostrally-innervated muscles.”

Figure 5: Part D. Readers need a key to the hatching. This is never explained adequately here. Also, have you explored displaying this data after a Fisher-Z transform of the correlations?

We revised the explanation for hatching in the Results section and legend of Figure 5D as below. We have not performed a Fisher-Z transform.

Lines 360-362

“most muscular conditions showed significant positive correlations between the magnitudes of PStEs and background EMGs (hatched bars in Figure 5D)”

Lines 378-381

“Hatched bars indicate the number of muscular conditions showing significant correlation between the magnitudes of PStEs and background EMGs (two-sided Pearson’s correlation test, *P* < 0.05). Unhatched bars show the conditions with no statistical significance.”

Figure 6 – same hatching issues – no description provided.

We revised the explanation for hatching in the legend of Figure 6E as follows:

Line 431

“Hatched and unhatched bars indicate the number of muscular conditions as in Figure 5D.”

Line 505-506 There should be citations here also of the work of Drs Takei and Seki.

The papers by Drs. Takei and Seki did not investigated directional tuning of activity of spinal interneurons, thus, we did not include their papers.

Line 546 Should potential current spread recruiting dorsal columns be discussed here?

We consider that the disappearance of Spinal PD (Figure 3A bottom panel) or no correlation between the magnitudes of background EMG and stimulus-induced muscle responses (Figure 6E, bottom-left panel) are due to current spread to the ventral aspect of the spinal cord. However, such a higher current will also recruit ascending pathways running on the dorsal columns in addition to dorsal rootlets. Thus, we revised the discussion as follows:

Lines 607-609

“These results indicate that current spread to the ventral aspect of the spinal cord leads to direct activation of motor axons, as well as recruitment of ascending pathways in the dorsal columns and dorsal rootlets.”

Line 569 The Ib recruitment explanation here seems naive. Couldn't various inhibitory pathways be more subliminal and 'in reserve' in normal voluntary control until stronger activity excites them and more Ia (and Ib) inputs combine?

We revised the discussion as follows:

Lines 632-644

“As shown in our results of onset latency (Figure 8D), facilitation effects were observed at as weak intensity as suppression effects. However, results showed suppressed voluntary torques at lower currents (Figure 7B), indicating stronger suppression effects via inhibitory interneurons. The neural mechanisms underlying the suppressed voluntary torques are unclear, but afferent inputs mediated by Ia presynaptic inhibition to motoneurons or autogenic inhibition to agonist motoneurons via inhibitory interneurons might be possible mechanisms for suppression effects at lower currents. The inhibitory influence mediated by presynaptic mechanisms (Eccles et al., 1961, 1962; Rudomin and Schmidt, 1999) are known to act on afferents, but not corticospinal inputs to motoneurons (Jackson et al., 2006). In addition, during muscular contraction, autogenic inhibition to agonist motoneurons via inhibitory interneurons is driven by Ib afferents (Houk, 1979; Lundberg and Malmgren, 1988). Our results suggest that lower currents predominantly result in these effects on the agonist motoneurons and suppress voluntary torque.”

Line 587 "Stimulations at most stimulus sites showed that the PDs for facilitation and suppression effects were similar to those for background EMGs (Figure 4G and H), indicating the trans-synaptic recruitment of motoneurons" … Please explain this better, unpack it for the reader.

We revised the explanation as follows:

Lines 695-701

“Stimulations at most stimulus sites showed that the PDs for facilitation and suppression effects were similar to those for background EMGs (Figures 4G and H). The correspondence in the PDs between the stimulus-induced muscle response and background EMGs indicates that changes in the amount of voluntary descending commands principally accounts for the torque direction-dependent modulation in the evoked muscle response. Such change in the evoked muscle response suggests that spinal stimulation produced trans-synaptic inputs to motoneurons that can be spatiotemporally summated by voluntary descending inputs.”

Line 609 There is no real optimization here, just a description of a band where both excitation and inhibition are coordinated.

We revised the term “optimized” as described above (responses to the sixth comment). We have replaced the term “optimized currents” by “the current of 150-1350 μA in the range where excitation and inhibition coordinated” on lines 728-729.

Line 780 – uncited reference to a 'V-test'.

We added the following citation:

Batschelet E. 1981. Circular Statistics in Biology. London: Academic Press.

Reviewer #2 (Recommendations for the authors):1. In my opinion, it would benefit the readability of the manuscript if the authors change the statement discussed in my 1st comment to "interaction of spinal stimulation with voluntarily evoked EMG and torque outputs" throughout the manuscript.

We aimed to systematically investigate how descending commands influence the spinal activity driven by spinal stimulation. To achieve this, we tried to capture the fundamental characteristics of stimulus-activated spinal circuits by probing the modulation of stimulus-evoked muscle responses and torques at each different stimulus intensity separately. Therefore, we would like to continue with our original objective and interpretation.

2. Why was the spinal stimulation ISI 197 ms, i.e., at approx. 5 Hz? Usually, therapeutic epidural stimulation is delivered at 50 Hz.

Since our study aimed to capture fundamental effects of descending commands on motor outputs induced by subdural spinal stimulation, we used stimulation at approximately 5 Hz. As seen in the results shown in Figure 1E, the evoked toque trajectory appeared at ~ 50-100 ms post-trigger and then returned to baseline within 150 ms after stimulation onset. Stimulation at 50 Hz would not have provided such a fundamental feature of motor output induced by spinal stimulation. We agree with the reviewer's opinion that therapeutic spinal stimulation is often delivered at 50 Hz. Thus, the spinal stimulation at higher frequency should be investigated in future studies to explore the therapeutic effect of spinal stimulation.

3. The authors do not state how many trials were performed per current level?

We added a summary of the collected data as Table 1 and 2 in Supplementary File 1. Also, we added the number of trials performed in experiments in the Materials and methods section as described below. Please refer the response to your 1st comment as well.

Lines: 823-824

“Each experiment consisted of 63-1004 successful trials (Table 2 in Supplementary File 1).”

Were these results tested for stability across days?

It took many days to get our data with various stimulus conditions as described in the response to your previous comments. Therefore, we did not test the motor outputs to spinal stimulation in the same stimulus condition across days. To investigate the stability of motor outputs, another study will be needed.

Given that voluntarily evoked activity could vary based on factors such as motivation, how does that affect the efficacy of spinal stimulation?

We used a juice reward to maintain the animals' motivation while they performed the task. Animals were required to maintain torque within the target for 0.7-0.8 s to receive a juice reward and were trained in this task for over 7 months. Finally, we collected data only from successful trials while the animals' motivation was maintained.

Although Reviewer 2 raised this very interesting question, the current study was not designed to investigate the factor of motivation on muscle responses by spinal stimulation. Our study on the motivational control of motor output, published in J Physiology (Suzuki et al., 2021), may provide some insight into your question.

4. The authors discuss the results in relation to polysynaptic inhibition and monosynaptic excitation that are present in the spinal circuits. I would highly encourage the authors to cite Guiho et al., JNE 2021 which proposes a potential novel model of spinal circuitry, and include it in the discussion.

We cited this paper for additional discussion as follows:

Lines 588-593

“In line with these considerations, Guiho et al. (2021) recently proposed a model of spinal circuitry driven by spinal electrical stimulation. In their model, the discharges of excitatory and inhibitory interneurons elicited by spinal stimulation are assumed to be integrated into motoneuron activity receiving corticospinal drives. The current study extends this model so that the voluntary descending commands are integrated with afferent inputs at spinal interneurons as well as motoneurons.”

Added citation

Guiho T, Baker SN, Jackson A. 2021. Epidural and transcutaneous spinal cord stimulation facilitates descending inputs to upper-limb motoneurons in monkeys. *J Neural Eng* 18:046011. doi:10.1088/1741-2552/abe358

Reviewer #3 (Recommendations for the authors):Overall, the manuscript is clear and compelling. Some modifications to the figures and text could help to improve clarity:1. In most figures, the PD vector uses the same color and line width as the rest of the polar plot. It would be helpful to highlight the PD vector by making the line thicker or a slightly different color.

We changed the color and thickness of the PD vectors in Figures 2D, 3E, 4G, 4H, and 7F to highlight the vectors.

2. It is not currently clear how different levels of background EMG activity were achieved. Were the monkeys trained to vary this activity level or were these differences analyzed post hoc based on the activity they happened to produce during various trials?

The reviewer’s latter interpretation is correct. We did not prepare the task in which the monkey had to produce different magnitudes of wrist torque to track the target. We classified individual stimulus data based on the pre-stimulus background EMG level throughout the experiments. We added a description how different levels of background EMG activity were achieved in the Results section.

Lines 354-356

“EMG activity obtained from the 8-directional task was divided into 5 different levels of background EMGs for analysis, and PStEs were shown based on the magnitudes of background EMG (Figure 5A-C).”

3. In Figure 5, I think the plots show increasing background activity moving from left to right, but this isn't currently labeled in the figure.

We added labels to show increasing background activity moving from left to right in Figures 5 and 6.

4. The descriptions of Figure 7 are confusing (e.g. center of peripheral panels, outer peripheral panels). It might be helpful to add an additional panel to the figure that shows the structure of each of the other panels without any data but including labels to help clarify what the different parts of each panel are showing.

We added a panel that shows the structure of each of the other panels in Figure 7A.

The legend of Figure 7A

Lines: 483-487

“(A) The structure of Figures 7B-D. The center panel shows evoked torque trajectory about wrist joint (top) and evoked EMG (bottom) during the hold period for the center target. Inner- (gray circles) and outer-peripheral panels (black circles) indicate evoked torque trajectory and evoked EMG on the 8-peripheral targets, respectively.”

5. It would be helpful to redefine the abbreviations for muscles in the methods section (i.e. at lines 670-672).

We added the abbreviations for muscles in the Materials and methods section.

Lines 807-813

“Three elbow muscles (biceps brachii [BB], brachioradialis [BR] and triceps brachii [Triceps]), six wrist muscles (pronator teres [PT], flexor carpi radialis [FCR], palmaris longus [PL], flexor carpi ulnaris [FCU], extensor carpi ulnaris [ECU] and extensor carpi radialis [ECR]), five digit muscles (flexor digitorum superficialis [FDS], flexor digitorum profundus [FDP], extensor digitorum communis [EDC], extensor digitorum 4 and 5 [ED4, 5] and abductor pollicis longus [APL]), and two intrinsic hand muscles (first adductor pollicis [ADP] and abductor digiti minimi [ADM]).”